# A shared inflammatory signature across severe malaria syndromes manifested by transcriptomic, proteomic and metabolomic analyses

Rafal S. Sobota [1,2], Emily M. Stucke[1], Drissa Coulibaly[3], Jonathan G. Lawton[1], Bryan E. Cummings[1], Savy Sebastian[1], Antoine Dara[3], James B. Munro[4], Amed Ouattara [1], Abdoulaye K. Kone[3], Bourama Kane[3], Karim Traoré [3], Bouréima Guindo[3], Bourama M. Tangara[3], Amadou Niangaly[3], Noah T. Ventimiglia[1], Modibo Daou[3,7], Issa Diarra[3], Youssouf Tolo[3], Mody Sissoko[3,8], Fayçal Maiga[3], Aichatou Diawara[3], Amidou Traore[3], Ali Thera[3], Matthew B. Laurens [1], Kirsten E. Lyke [1], Bourema Kouriba[3], Ogobara K. Doumbo[3,9], Christopher V. Plowe[1], David R. Goodlett [5], Joana C. Silva [4], Mahamadou A. Thera [3,6] & Mark A. Travassos [1,6] ✉

Factors governing the clinical trajectory of *Plasmodium falciparum* infection remain an important area of investigation. Here we present transcriptomic, proteomic and metabolomic analyses comparing clinical subtypes of severe *Plasmodium falciparum* malaria to matched controls with uncomplicated disease in 79 children from Mali. *MMP8*, *IL1R2*, and *ARG1* transcription is higher across cerebral malaria, severe malarial anemia, and concurrent cerebral malaria and severe malarial anemia, indicating a shared inflammatory signature. Tissue inhibitor of metalloproteinases 1 is the most upregulated protein in cerebral malaria, which along with elevated *MMP8* and *MMP9* transcription, underscores the importance of the metalloproteinase pathway in central nervous system pathophysiology. L-arginine metabolites are decreased in cerebral malaria, which coupled with increased *ARG1* transcription suggests a putative mechanism impairing cerebral vasodilation. Using multi-omics approaches, we thus describe the inflammatory cascade in severe malaria syndromes, and identify potential therapeutic targets and biological markers.

Malaria continues to cause significant morbidity and mortality in Africa. According to WHO estimates for 2023, 96% of the 597,000 malaria-related deaths globally occurred in Africa[1]. Severe malaria (SM) is most frequently caused by *Plasmodium falciparum*[1]. The most lethal subtypes of SM include cerebral malaria (CM), severe malarial anemia (SMA), respiratory distress, and combinations of these phenotypes[2].

SM burden is compounded by long-term sequelae. CM survivors have an increased risk of epilepsy, neurocognitive impairment, focal neurological deficits, and behavioral problems following infection[3,4]. SMA has also been associated with long-term neurocognitive impairment[5]. The pathophysiology underlying the development of various SM subtypes remains an active area of research.

Advances in high-throughput technology have facilitated characterization of transcriptomic, proteomic, and metabolomic signatures of biological processes that mediate the interaction between *P. falciparum* and the human host, but severe malarial subtypes in sub-Saharan Africa have not been comprehensively profiled in this manner[6–11]. To date, there are no published studies of severe malaria that describe transcriptomics, proteomics, and metabolomics in overlapping samples.

Here, we present an integrated analysis using these approaches for children with CM, providing additional insight by contextualizing the findings of each approach within the greater biological framework. In addition, we perform transcriptomic analyses of SMA, concurrent CM and SMA, and uncomplicated malaria without a history of cerebral malaria. We hypothesize that transcriptomic, proteomic, and metabolomic profiles of children with SM subtypes differ from each other and from children with uncomplicated disease.

## Results

### Differential gene expression in cerebral malaria
Twelve matched pairs met the inclusion criteria for comparing cases of CM to uncomplicated malaria controls without a history of CM. CM cases experienced greater parasitemia (126,603 versus 33,408 parasites/μL, respectively, *p*-value = 0.02; Table 1). There was no statistically significant difference in hemoglobin levels for the comparison.

Transcripts of *IL1R2, FKBP5*, and *MMP8* had the most significant association to CM in comparison to uncomplicated controls (logFC 4.41, 2.58, and 4.20, with FDR-adjusted *p*-values of 1.75E-11, 2.00E-09, and 2.73E-08, respectively; Fig. 1, Supplementary Table 1). 405 genes met the threshold for statistical significance in the comparison (Supplementary Data 1).

Gene Ontology (GO) pathway analyses revealed that T cell activation, positive regulation of cytokine production, and regulation of immune effector process were the top three biological processes for differentially expressed transcripts between CM and uncomplicated controls (Fig. 1D, Supplementary Table 2). Regulation of actin cytoskeleton and focal adhesion were among the eight significantly upregulated KEGG pathways for this comparison (Supplementary Table 3). No KEGG pathways were significantly downregulated in this comparison.

### Differential gene expression in severe malarial anemia
Eight matched pairs met the inclusion criteria in comparing cases of SMA to controls with uncomplicated disease. Hemoglobin levels were significantly lower in SMA compared to controls (2.87 versus 8.41 g/dL, respectively, *p*-value = 1.05E-04; Table 1). There was no statistically significant difference in parasitemia for the comparison.

Transcripts of *DAAM2, LTF*, and *ARG1* had the most significant associations with SMA in comparison to uncomplicated controls (logFC 4.83, 3.87, and 2.82, with FDR-adjusted *p*-values of 1.54E-03, 2.36E-03, and 4.97E-03, respectively; Fig. 2, Supplementary Table 4). 19 genes met the threshold for statistical significance in this comparison.

Gene Ontology (GO) pathway analyses revealed that negative regulation of cytokine production, tumor necrosis factor production, and killing of cells of other organisms were among the top biological processes for differentially expressed transcripts between SMA and controls (Fig. 2D, Supplementary Table 5). KEGG analyses showed that vitamin B6 metabolism, arachidonic acid metabolism, and metabolic pathways were among the 12 significantly upregulated pathways in comparing SMA to controls (Supplementary Table 6). No downregulated pathways met the threshold for significance in this comparison.

### Differential gene expression in concurrent cerebral malaria and severe malarial anemia
Eight matched pairs met the inclusion criteria for comparing cases of concurrent CM and SMA to uncomplicated controls without a history of CM. Concurrent CM and SMA cases had significantly lower hemoglobin levels compared to controls (3.30 versus 8.11 g/dL, respectively, *p*-value = 2.15E-03; Table 1). There was no statistically significant difference in parasitemia in this comparison.

Transcripts of *OLMF4, XAF1*, and *IL1R2* had the most significant association with concurrent CM and SMA in comparison to uncomplicated controls (logFC 6.90, −2.39, and 4.04, with FDR-adjusted *p*-values of 1.75E-05, 2.77E-03, and 2.77E-03, respectively; Fig. 3, Supplementary Table 7). 67 genes met the threshold for statistical significance in the comparison.

Gene Ontology (GO) pathway analyses revealed that defense response to a symbiont, response to type 1 interferon, and cytokine-mediated signaling pathway were among the top ten biological processes for the comparison of cases with concurrent CM and SMA to uncomplicated controls (Fig. 3D, Supplementary Table 8). KEGG analysis revealed that ubiquitin-mediated proteolysis was among the three statistically significant upregulated pathways, and necroptosis was significantly downregulated when comparing concurrent CM and SMA to controls (Supplementary Table 9).

### Differential gene expression between controls with and without a history of cerebral malaria
Sixteen matched pairs met the inclusion criteria in comparing controls with and without a history of CM. There was no statistically significant difference in age, hemoglobin levels, or parasitemia between controls

**Table 1 | Demographic characteristics for study participants used for transcriptomic analyses comparing severe malaria subtypes to uncomplicated controls without a history of cerebral malaria**

|  | Cerebral malaria | | | Severe malarial anemia | | | Concurrent CM and SMA | | |
|---|---|---|---|---|---|---|---|---|---|
|  | Cases | Controls | *p*-value | Cases | Controls | *p*-value | Cases | Controls | *p*-value |
| *n* | 12 | 12 |  | 8 | 8 |  | 8 | 8 |  |
| Female | 5 | 5 |  | 6 | 6 |  | 2 | 2 |  |
| Age* (yrs) | 2.63 (1.33) | 2.78 (1.24) | 0.39 | 2.50 (0.76) | 2.75 (0.71) | 0.17 | 3.25 (1.39) | 3.25 (1.28) | 1 |
| Parasitemia* (parasite/μL) | 126,603 (145,769) | 33408 (48,930) | 0.02 | 48010 (59,282) | 37237 (39,252) | 0.69 | 64,125 (95,799) | 9928 (11,418) | 0.15 |
| BCS |  |  |  |  |  |  |  |  |  |
| 5 | 0 | 12 |  | 4 | 8 |  | 0 | 8 |  |
| 4 | 0 | 0 |  | 1 | 0 |  | 0 | 0 |  |
| 3 | 0 | 0 |  | 3 | 0 |  | 0 | 0 |  |
| 2 | 9 | 0 |  | 0 | 0 |  | 4 | 0 |  |
| 1 | 3 | 0 |  | 0 | 0 |  | 4 | 0 |  |
| Hgb* (g/dL) | 7.68 (1.69) | 8.87 (1.72) | 0.12 | 2.87 (0.55) | 8.41 (1.83) | 1.05E-04 | 3.30 (0.91) | 8.11 (2.43) | 2.15E-03 |

*denotes mean (standard deviation). Statistical significance was assessed using a two-tailed *t*-test.

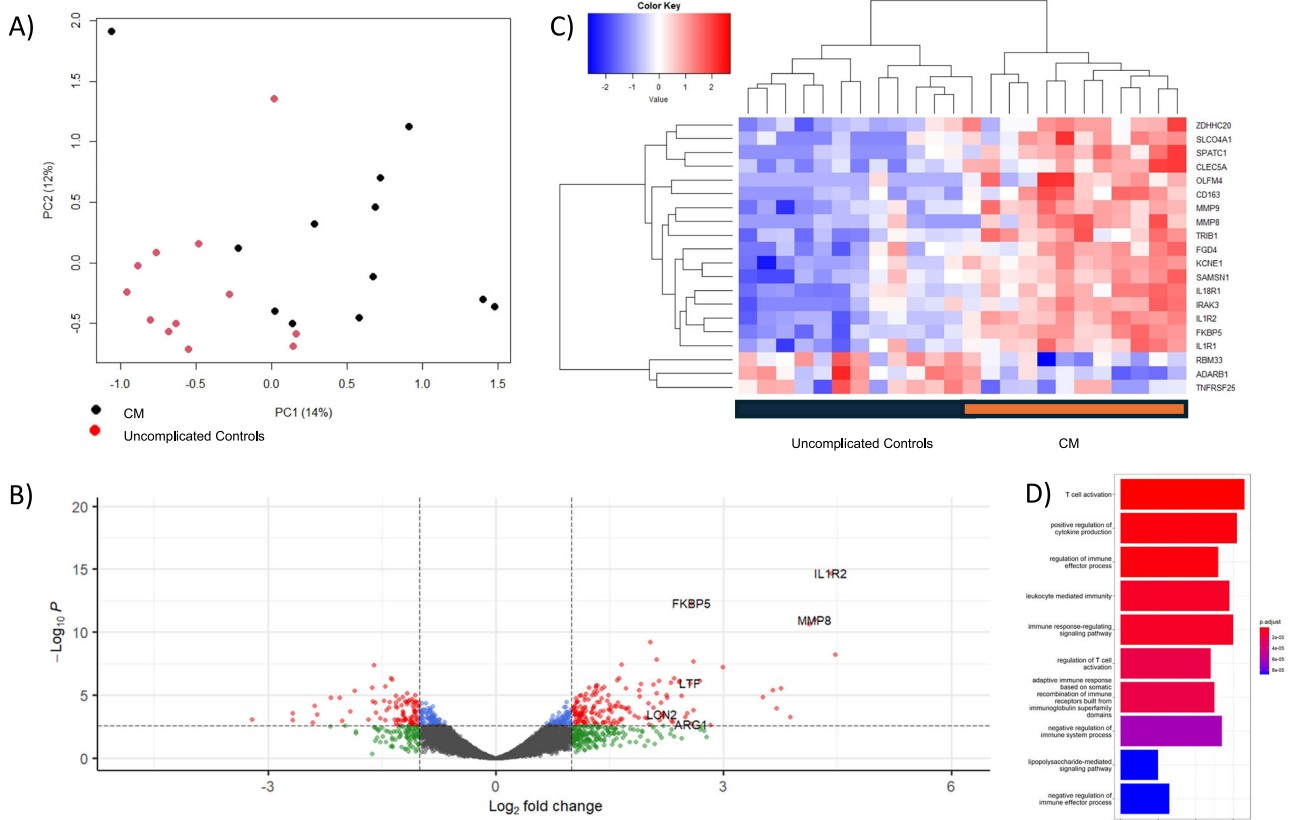

**Fig. 1 | Differential gene expression analysis comparing cerebral malaria to uncomplicated controls without a history of cerebral malaria. A** Principal component analysis showing the separation of CM from UM along the first two principal components. **B** Volcano plot showing the significance of association from a quasi-likelihood *F* test based on the negative binomial distribution (red and blue FDR-adjusted *p*-value < 0.05) and effect size (green and red >1 absolute log₂-fold change). **C** Heatmap of the top 20 differentially expressed transcripts, showing the clustering of associated genes from a quasi-likelihood *F* test (y-axis) and CM cases versus UM controls (x-axis). **D** Bar plot of the top 10 significantly associated gene ontology biological processes. Enrichment analysis was performed using the clusterProfiler package in R. Enriched terms were determined using a hypergeometric test, with *p*-values < 0.05 considered significant. Pathway names are represented on the Y-axis, and gene counts on the X-axis. The color spectrum represents the extent of statistical association, from red, *p*-value 1E-05, to blue, *p*-value 1E-04.

with and without a history of CM (Supplementary Table 10). There were no significantly associated transcripts at the FDR corrected level for differential gene expression analysis in comparing controls with and without a history of CM.

**Differential gene expression between severe malaria subtypes**

21 cases of CM to 20 cases of SMA met the inclusion criteria for an unpaired comparison of gene expression. CM cases had significantly higher hemoglobin (7.62 versus 3.56 g/dL, *p*-value 1.8E-12) and parasitemia levels (145,634 versus 61,859 parasites/μL, *p*-value 0.022; Table 2). There was no statistically significant difference in gender or age. There were 165 differentially expressed genes between the two groups, 137 of which were expressed at higher levels in SMA. *H1-2*, *H1-0*, and *SCL2A1* were the transcripts with the most significant association for genes expressed at a higher level in SMA (logFC 1.86, 2.37, and 2.96; *p*-values 4.15E-04, 1.55E-03, and 1.55E-03, respectively; Fig. 4, Supplementary Table 11). *GBP3*, *H2BC21*, and *SELENOT* were the transcripts with the most significant association for genes expressed at a higher level in CM (logFC 1.26, 1.31, and 0.79; *p*-values 8.06E-03, 8.21E-03, and 9.47E-03, respectively; Supplementary Table 12). Gene Ontology (GO) pathway analyses revealed that cellular response to toxic substance and erythrocyte differentiation were among the top biological processes for the comparison of CM to SMA (Fig. 4D, Supplementary Table 13). KEGG analyses revealed that GABAergic synapse and cortisol synthesis and secretion were among the significantly associated pathways upregulated in SMA, while taurine and

hypotaurine metabolism were the only significantly associated pathways upregulated in CM (Supplementary Table 14).

21 cases of CM and 10 cases of concurrent CM and SMA met the inclusion criteria for an unpaired comparison of gene expression. CM cases had significantly higher hemoglobin levels (7.62 versus 3.21 g/dL, *p*-value 1.3E-08; Table 2). There were no statistically significant differences in parasitemia, gender, or age. Unpaired analysis comparing cases of CM to cases of concurrent CM and SMA yielded 19 differentially expressed genes between the two groups, four of which were higher in CM. *MPO*, *TBC1D14*, and *ELANE* were among the transcripts with the most significant association for genes expressed at a higher level in concurrent CM and SMA (logFC 3.75, 2.77, and 3.16; *p*-values 2.03E-03, 6.58E-03, and 0.020, respectively; Supplementary Fig. 1, Supplementary Table 15). *SUB1*, *SSR3*, *TMED2*, and *JCHAIN* were the transcripts with the most significant association for genes expressed at a higher level in CM (logFC 0.89, 1.27, 0.95, and 1.82; *p*-values 0.020, 0.028, 0.034, and 0.038, respectively; Supplementary Table 15). KEGG analyses revealed that TGF-beta signaling pathway, apelin signaling pathway, and steroid biosynthesis were among the significantly associated pathways upregulated in concurrent CM and SMA, while TNF signaling pathway and GnRH signaling pathway were among the significantly associated pathways upregulated in CM (Supplementary Table 16).

20 cases of SMA and 10 cases of concurrent CM and SMA met the inclusion criteria for an unpaired comparison of gene expression. There was no statistically significant difference in hemoglobin levels,

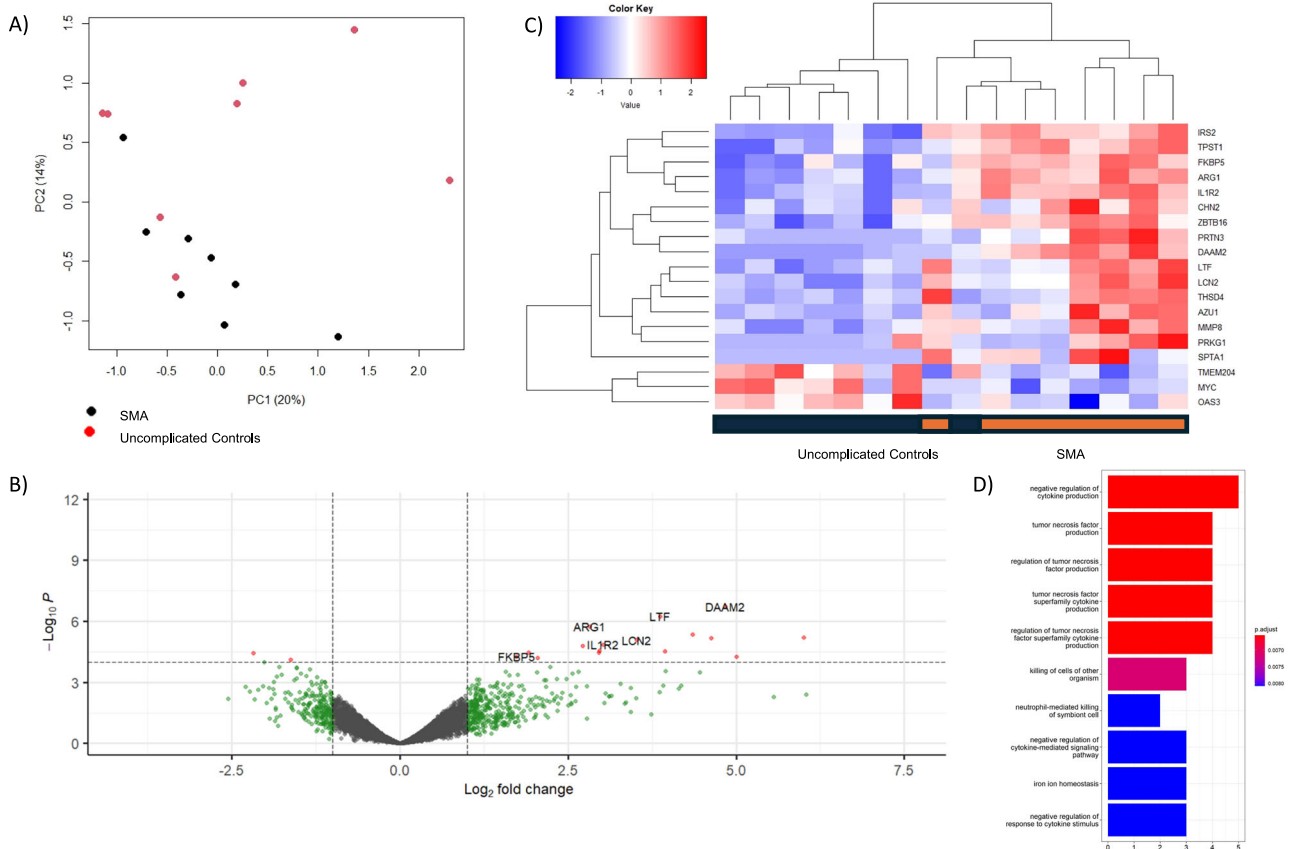

**Fig. 2 | Differential gene expression analysis comparing severe malarial anemia cases to uncomplicated controls. A** Principal component analysis showing the separation of SMA from UM along the first two principal components. **B** Volcano plot showing the significance of association from a quasi-likelihood *F* test based on the negative binomial distribution (red FDR-adjusted *p*-value < 0.05) and effect size (green and red >1 absolute log₂-fold change). **C** Heatmap of the differentially expressed transcripts, showing the clustering of associated genes from a quasi-

likelihood *F* test (y-axis) and CM cases versus UM controls (x-axis). **D** Bar plot of the top 10 significantly associated gene ontology biological processes. Enrichment analysis was performed using the clusterProfiler package in R. Enriched terms were determined using a hypergeometric test, with *p*-values < 0.05 considered significant. Pathway names are represented on the Y-axis, and gene counts on the X-axis. Color spectrum represents the extent of statistical association, from red, *p*-value 6.5E-03, to blue, *p*-value 8E-03.

parasitemia, sex, or age (Table 2). Unpaired analysis comparing cases of SMA to cases of concurrent CM and SMA yielded four differentially expressed genes between the two groups, all of which were higher in the concurrent CM and SMA group; GNA13, CREBBP, and WDR74 were among the transcripts with the most significant association for genes expressed at a higher level in concurrent CM and SMA (logFC 3.82, 3.02, and 3.67; *p*-values 0.025, 0.031, and 0.049, respectively; Supplementary Table 17). KEGG analyses revealed that TGF-beta signaling pathway and steroid biosynthesis were among the significantly associated pathways upregulated in concurrent CM and SMA (Supplementary Table 18).

**Proteomics in cerebral malaria**
Serum proteins were assayed for 14 age-, sex-, and ethnicity-matched pairs of CM and uncomplicated malaria controls without history of CM, 11 of which overlapped with the transcriptomic analyses. CM cases had higher average parasitemia (107,330 versus 28,670 parasites/μL, *p*-value 0.039), and there was no statistically significant difference in hemoglobin (Supplementary Table 19). After removing data with ambiguous assignments, 180 of 212 proteins remained for comparison, 43 of which were statistically significant (Fig. 5A, Supplementary Tables 20 and 21). Of the 28 proteins that were higher in CM, TIMP1 was the most significant (logFC 1.80, *p*-value 1.71E-03). Of note, CD14, VCAM-1, and ICAM-1 levels were also significantly higher in CM cases (Supplementary Table 20). SERPIND1 had the most significant

association for proteins with lower levels in CM than controls (logFC −0.55, *p*-value 1.65E-03). APOM, C3, and C1QB were also among the proteins expressed at significantly lower levels in CM (Supplementary Table 21). Response to hypoxia and response to endogenous stimulus were among the most significantly upregulated GO biological processes in CM (Supplementary Table 22). Antimicrobial humoral immune response mediated by antimicrobial peptide, granulocyte chemotaxis, positive regulation of TGFβ1 production, and positive regulation of epithelial cell apoptotic process were significantly downregulated in CM (Supplementary Table 23). Kegg pathway analysis revealed that the complement and coagulation cascades were the only statistically significant pathways, and it was downregulated in CM.

Of the 180 assayed proteins, 91 had sequenced transcripts that passed quality controls in the CM comparison to uncomplicated controls. Six of the 405 transcripts that were significantly associated with CM to uncomplicated controls had protein level data. CD14 was the only significantly upregulated transcript in CM (logFC 0.85, *p*-value 0.044) that was also significantly upregulated in the proteomic analysis (logFC 0.64, *p*-value 2.00E-03).

**Metabolomics in cerebral malaria**
Metabolomic profiles were ascertained for the same set of 14 CM cases and 14 age-, sex-, and ethnicity-matched uncomplicated malaria controls without a history of CM as in the proteomic analysis. After removing ambiguous assignments, 147 lipids and metabolites met the

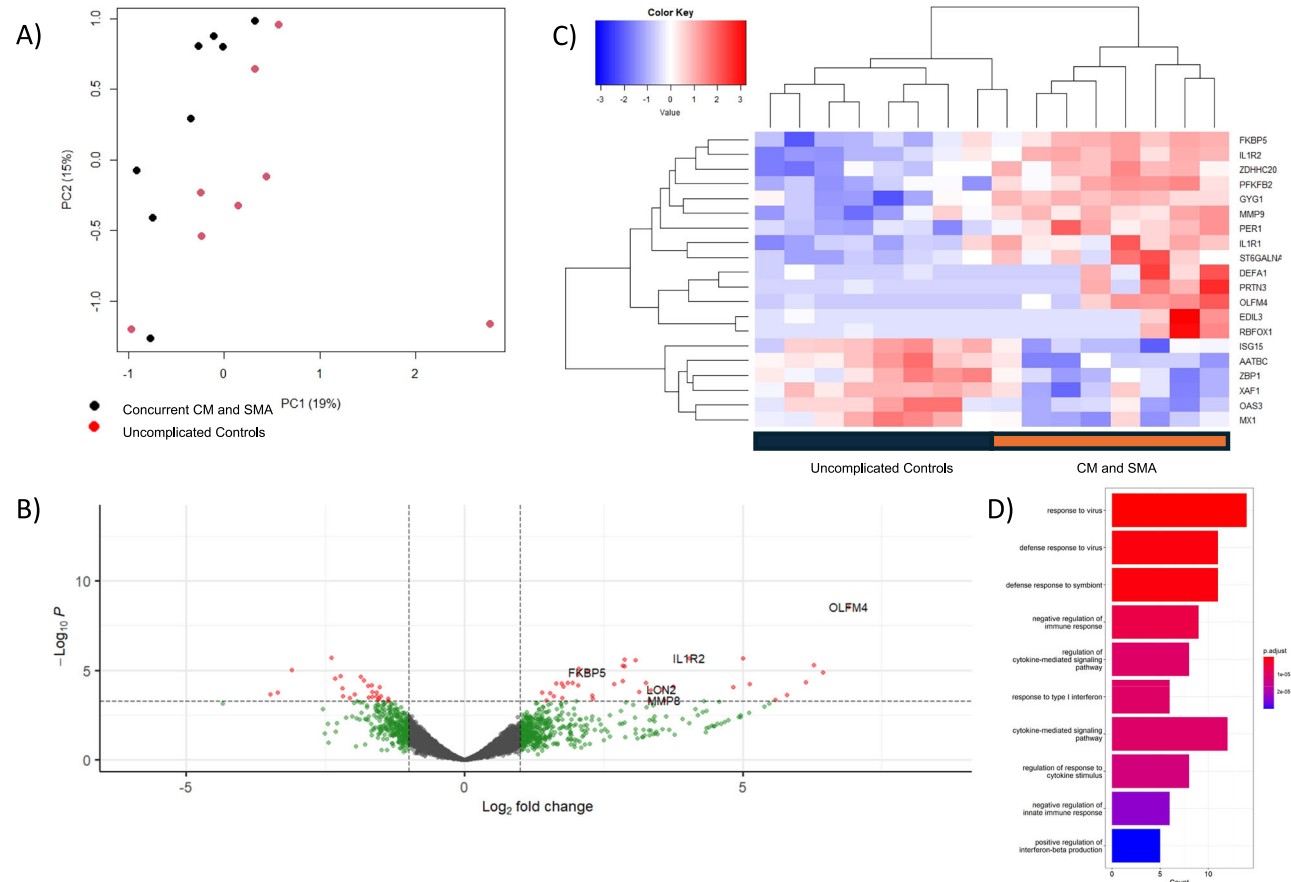

**Fig. 3 | Differential gene expression analysis comparing concurrent cerebral malaria and severe malarial anemia to uncomplicated controls without a history of cerebral malaria. A** Principal component analysis showing the separation of concurrent CM and SMA from UM along the first two principal components. **B** Volcano plot showing the significance of association from a quasi-likelihood *F* test based on the negative binomial distribution (red FDR-adjusted *p*-value < 0.05) and effect size (green and red >1 absolute log₂-fold change). **C** Heatmap of the top 20 differentially expressed transcripts, showing the clustering of associated genes from a quasi-likelihood *F* test (y-axis) and CM cases versus UM controls (x-axis). **D** Bar plot of the top 10 significantly associated gene ontology biological processes. Enrichment analysis was performed using the clusterProfiler package in R. Enriched terms were determined using a hypergeometric test, with *p*-values < 0.05 considered significant. Pathway names are represented on the Y-axis, and gene counts on the X-axis. Color spectrum represents the extent of statistical association, from red, *p*-value 9E-06, to blue, *p*-value 3E-05.

**Table 2 | Demographic characteristics for study participants used in severe malaria subtype comparisons**

|  | CM | SMA | *p*-value | CM | Concurrent | *p*-value | SMA | Concurrent | *p*-value |
|---|---|---|---|---|---|---|---|---|---|
| *n* | 21 | 20 |  | 21 | 10 |  | 20 | 10 |  |
| Female | 10 | 10 | 0.88 | 10 | 2 | 0.14 | 10 | 2 | 0.11 |
| Age* (yrs) | 3.12 (1.34) | 2.72 (1.46) | 0.36 | 3.12 (1.34) | 3.10 (1.29) | 0.96 | 2.72 (1.46) | 3.10 (1.29) | 0.49 |
| Parasitemia* (parasite/μL) | 145,634 (143,113) | 61,859 (66,559) | 0.022 | 145,634 (143,113) | 51,367 (88,664) | 0.067 | 61,859 (66,559) | 51,367 (88,664) | 0.72 |
| BCS |  |  |  |  |  |  |  |  |  |
| 5 | 0 | 14 |  | 0 | 0 |  | 14 | 0 |  |
| 4 | 0 | 1 |  | 0 | 0 |  | 1 | 0 |  |
| 3 | 0 | 5 |  | 0 | 0 |  | 5 | 0 |  |
| 2 | 10 | 0 |  | 10 | 5 |  | 0 | 5 |  |
| 1 | 11 | 0 |  | 11 | 5 |  | 0 | 5 |  |
| Hgb* (g/dL) | 7.62 (1.62) | 3.56 (0.76) | 1.8E-12 | 7.62 (1.62) | 3.21 (1.05) | 1.3E-08 | 3.56 (0.76) | 3.21 (1.05) | 0.3 |

*denotes mean (standard deviation). Statistical significance was assessed using the chi-squared for sex and a two-tailed *t*-test for age, parasitemia, and hemoglobin.

multiple testing-corrected level of significance. Nervonic acid (logFC 0.89, *p*-value 0.026), pipecolic acid (logFC 1.76, *p*-value 0.044), cortisol (logFC 1.59, *p*-value 3.61E-03), and mannitol (logFC 7.43, *p*-value 0.014) were among the metabolites with significantly higher levels in CM (Fig. 5B, Supplementary Table 24, Supplementary Data 3).

Paracetamol was also higher in CM cases (logFC 7.37, *p*-value 0.035). Arginine and linoleic acid were among the metabolites with lower levels in cases of CM compared to uncomplicated controls (logFC −1.04 and −0.79, *p*-value 0.038 and 0.017, respectively; Supplementary Table 25). Pathway analysis of serum metabolites in CM compared to

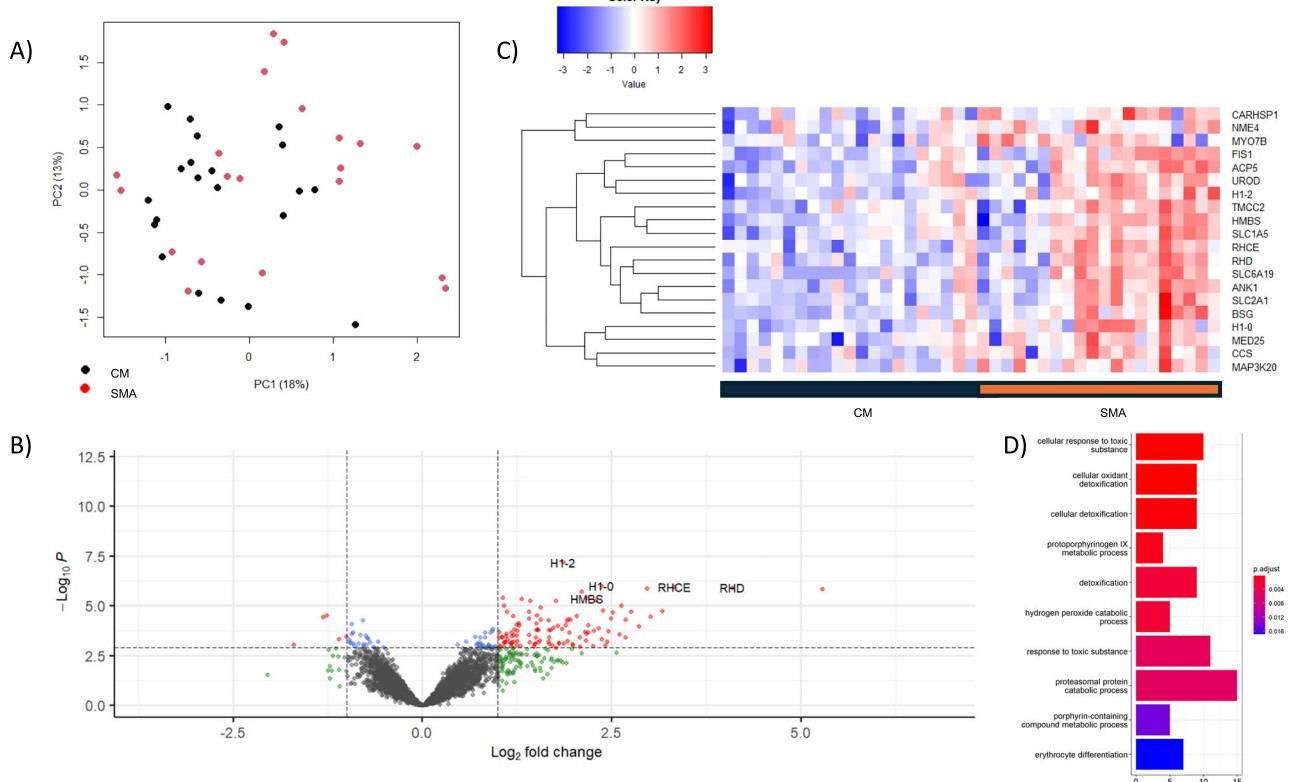

**Fig. 4 | Differential gene expression analysis comparing cerebral malaria to severe malarial anemia. A** Principal component analysis showing the separation of CM from SMA along the first two principal components. **B** Volcano plot showing the significance of association from a quasi-likelihood $F$ test based on the negative binomial distribution (red and blue FDR-adjusted $p$-value < 0.05) and effect size (green and red >1 absolute $log_2$-fold change). **C** Heatmap of the top 20 differentially expressed transcripts showing the clustering of associated genes from a quasi-likelihood $F$ test (y-axis). **D** Bar plot of the top 10 significantly associated gene ontology biological processes. Enrichment analysis was performed using the clusterProfiler package in R. Enriched terms were determined using a hypergeometric test, with $p$-values < 0.05 considered significant. Pathway names are represented on the Y-axis, and gene counts on the X-axis. The color spectrum represents the extent of statistical association, from red, $p$-value 2E-03, to blue, $p$-value 0.018.

uncomplicated malaria controls revealed that phenylalanine, tyrosine, and tryptophan biosynthesis, and alanine, aspartate, and glutamate metabolism were among significantly upregulated pathways in cerebral disease (Supplementary Table 26).

### Differential gene expression in cerebral malaria compared to controls with a history of cerebral malaria

Eight matched pairs met the inclusion criteria for comparing CM cases to uncomplicated malaria controls with a history of CM. Parasitemia was not significantly higher in CM cases than controls with a history of CM (189,211 versus 37,837 parasites/µL, respectively, $p$-value = 0.10; Supplementary Table 27A). There was no statistically significant difference in hemoglobin levels for the comparison.

Transcripts of *IL1R2, CD163,* and *FKBP5* had the most significant association in the comparison between CM and controls with a history of CM (logFC 4.13, 3.36, and 2.18 with FDR-adjusted $p$-values of 1.51E-05, 2.77E-04, and 8.34E-04, respectively; Supplementary Fig. 3A). 72 genes met the threshold for statistical significance in this comparison (Top 20 are listed in Supplementary Table 28). 57 statistically significant transcripts overlapped in comparisons of CM to controls with and without a history of CM. *IL10* was the most significantly upregulated gene in the comparison of CM to uncomplicated controls with a history of CM (logFC 3.49 and FDR-adjusted $p$-value 6.12E-03) that did not associate with uncomplicated controls without a history of CM (logFC 1.16 and FDR-adjusted $p$-value 0.12).

Response to hydrogen peroxide, cellular response to reactive oxygen species, and cellular response to chemical stress were among

the top ten biological pathways in GO analysis when comparing CM to controls with a history of CM (Supplementary Fig. 4A).

### Differential gene expression in severe malarial anemia compared to controls with a history of cerebral malaria

Six matched pairs met the inclusion criteria in comparing cases of SMA to controls with a history of CM. Hemoglobin levels were significantly lower in SMA compared to controls with a history of CM (3.35 versus 9.45 g/dL, respectively, $p$-value 5.27E-05; Supplementary Table 27B). There was no statistically significant difference in parasitemia for the comparison.

No genes reached the threshold for statistical significance when comparing SMA to controls with a history of CM.

### Differential gene expression in concurrent cerebral malaria and severe malarial anemia versus controls with a history of cerebral malaria

Four matched pairs met the inclusion criteria for comparing cases of concurrent CM and SMA to controls with a history of CM. Concurrent CM and SMA cases had significantly lower hemoglobin levels compared to controls without and with a history of CM (2.97 versus 8.77 g/dL, respectively, $p$-value 0.023; Supplementary Table 27C). There was no statistically significant difference in parasitemia in this comparison.

Transcripts of *MPO, IFI44L*, and *ELANE* had the most significant association in the comparison between concurrent CM and SMA to controls with a history of CM (logFC 8.77, −6.49, and 9.93 with FDR-adjusted $p$-values of 3.52E-05, 1.57E-04, and 1.57E-04, respectively;

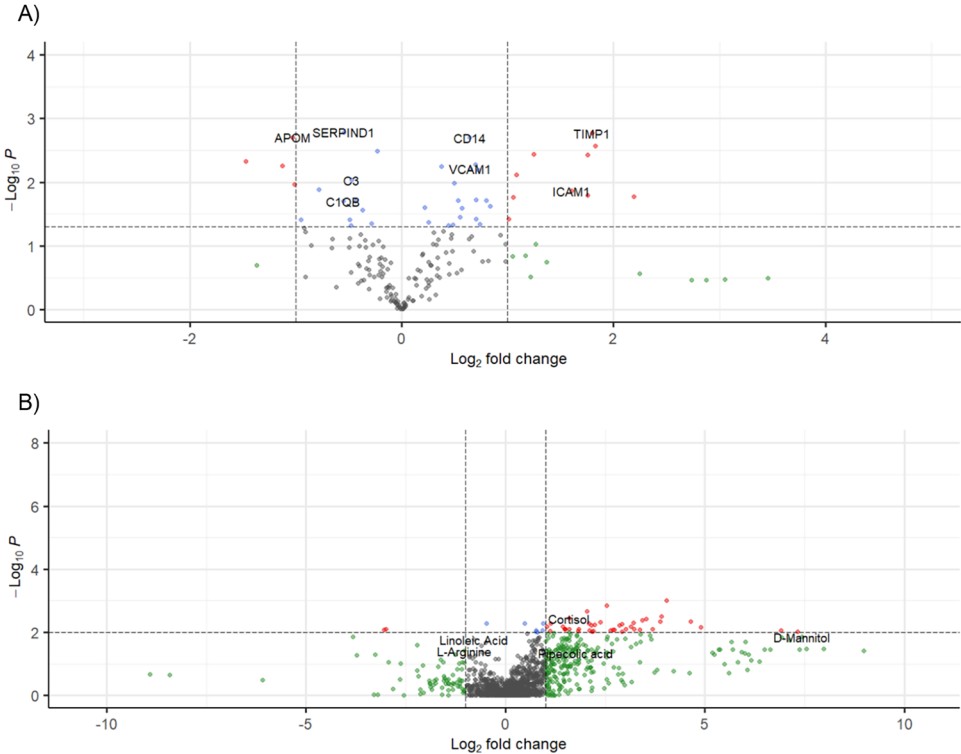

**Fig. 5 | Proteomic and metabolomic comparison of cerebral malaria to uncomplicated controls without a history of cerebral malaria.** Volcano plot showing significance of association assessed with a two-tailed paired *t*-test (red and blue FDR-adjusted *p*-value < 0.05) and effect size (green and red >1 absolute log₂-fold change) in comparisons of cerebral malaria to uncomplicated malaria without a history of cerebral malaria in **A** proteomic and **B** metabolomics analyses.

Supplementary Fig. 3B). 67 genes met the threshold for statistical significance in this comparison (Top 20 transcripts are listed in Supplementary Table 29). 26 statistically significant transcripts overlapped in comparisons of concurrent CM and SMA to controls with and without a history of CM.

Gene Ontology (GO) pathway analyses revealed that defense response to a symbiont, response to type 1 interferon, and regulation of cytokine-mediated signaling pathway were among the top ten biological processes for the comparison of cases with concurrent CM and SMA to controls with a history of CM (Supplementary Fig. 4B).

### Deconvolution

Deconvolution fractions of T cells, B cells, and neutrophils did not significantly associate with any of the severe malaria outcomes when compared to controls without a history of CM, and only SMA compared to T cell proportion was statistically significant in comparisons to controls with a history of CM (*p*-value 5.50E-03, Supplementary Table 30). Principal component analyses for comparisons of severe malaria subtypes to controls without a history of CM with and without adjustment for deconvolution are presented in Supplemental Fig. 5A–F. Adjustment for deconvolution lowered the number of genes passing the FDR threshold from 405 to 145 in the CM versus controls without a history comparison, and the top three associated genes without adjustment were significant after adjusting for B cell, T cell, and neutrophil proportions (Supplementary Table 31). Adjustment for deconvolution lowered the number of significantly associated genes from 19 to 9 for the SMA versus controls without a history comparison, and the top three genes associated without adjustment for cell proportions remained significant (Supplementary Table 32). Adjustment for deconvolution lowered the number of genes passing the FDR threshold from 67 to 20 for the concurrent CM and SMA versus controls without a history comparison, and two of the top three associating genes without the adjustment for cell proportions (*OLFM4* and *XAF1*) remained significant after (Supplementary Table 33).

Adjustment for deconvolution lowered the number of genes passing the FDR threshold from 72 to 1 in the CM versus controls with a history of CM comparison, and 67 to 43 for the concurrent CM and SMA versus controls with a history of CM comparison (Supplementary Table 34A, B, respectively). No genes were significant in comparing SMA to controls with a history of CM, with and without adjustment for deconvolution.

Deconvolution analyses adjusting for T cell, B cell, and neutrophil proportions did not significantly alter the clustering of samples, although it did lower the number of significant genes in each comparison. However, we believe that the robustness of our findings to different cell proportions further validates our findings.

### Discussion

We identified transcriptomic associations for CM, SMA, and concurrent CM and SMA and replicated findings from prior gene expression studies. Transcriptome analysis of severe malarial anemia is important given that it is the leading cause of death from malaria[2]. We found common signaling patterns across all SM subtypes, as well as transcripts that differentiated the phenotypes from each other. We also performed untargeted proteomic and metabolomic comparisons of sera from CM cases to controls without a history of CM, again discovering associations and validating prior findings. Using a paired approach, with most samples overlapping in the transcriptomic, proteomic, and metabolomic analyses, we comprehensively detailed features of the inflammatory cascade in CM.

Matrix metalloproteinases (MMPs) are a family of zinc-dependent endopeptidases that play an essential role in the extracellular matrix breakdown, including the BBB in the central nervous system[12,13]. Transcription of *MMP8* was increased in all SM subtypes in comparison to uncomplicated controls. *MMP8* encodes for matrix

metalloproteinase 8, a neutrophil secreted granule protein with collagenase activity previously associated with blood-brain barrier breakdown in malarial retinopathy and CM[14–16]. Genetic downregulation and pharmacologic inhibition of *MMP8* have been associated with neuroprotection in a murine sepsis model, where animals with elevated levels of the protein had increased central nervous system inflammation and toxicity[17]. *MMP9* encodes for a protein involved in the digestion of gelatin, the denatured form of collagen, and is released by neutrophils, endothelial cells, and astrocytes[18,19]. Activation of MMP9 leads to the digestion of type IV collagen, laminin, and fibronectin, which in turn leads to the breakdown of BBB tight junctions[20]. *MMP9* transcripts were significantly associated with CM and concurrent CM and SMA, suggesting that they might be specific for SM subsets that involve cerebral disease. Tissue inhibitor of metalloproteinases 1 (TIMP1) is known to protect the BBB by inhibition of active MMPs and has neuroprotective effects through dampening the effects of glutamate in excitotoxic injury[21]. TIMP1 has been shown to bind MMP9, and it is secreted by many tissues, including astrocytes in the central nervous system[22,23]. TIMP1 had the strongest association in cases of CM compared to uncomplicated controls in proteomic analyses, and it was upregulated in CM. TIMP1 was also upregulated in CM cases compared to uncomplicated controls in transcriptomic analyses (logFC 0.76), but the *p*-value was not significant after correcting for multiple testing (unadjusted 0.012, adjusted 0.11). Evaluation of serum levels of MMP8 and TIMP1 in children from Gabon has shown that TIMP1 was associated with signs and symptoms of severe malaria, whereas MMP8 was elevated in both severe malaria and unmatched mild malaria cases compared to healthy controls[24]. In this study, MMP8 and MMP9 were associated with SM subtypes compared to uncomplicated controls, but they were not differentially expressed in direct comparisons of SM subtypes with each other. Power for the latter analyses may have been limited due to sample sizes and a lack of matching.

*IL1R2* and *FKBP5* have not been previously associated with malaria. *IL1R2* encodes interleukin 1 receptor type 2, a decoy receptor for IL-1 that lacks a toll/interleukin receptor (TIR) domain[25]. This prevents the progression of an inflammatory cascade induced by IL-1 binding of toll-like receptors[26]. *IL1R2* expression is stimulated by glucocorticoids and IL-4, and the soluble form of the protein is produced by metalloprotease activity[27]. Polymorphisms of *IL1R2* have been previously associated with preterm delivery in African American mothers, suggesting a possible link between the observed association of malaria and preterm labor[28,29]. *FKBP5* encodes FK506 binding protein 5 (FKBP51), an immunophilin preferentially expressed in T cells and adipocytes[30]. *FKBP5* expression is induced by steroids, and the protein acts as a co-chaperone with Hsp90 to limit the ability of bound glucocorticoid receptors (GR) to induce transcription[31]. FKBP51 has been shown to reduce direct binding of steroids to GR and limit the nuclear translocation of activated GR into the nucleus[32]. Polymorphisms in *KFBP5* have been associated with major depressive disorder, PTSD, and bipolar disorder in humans, and *KFBP5*[−/−] mice were resistant to stress-induced depressive behavior, underscoring the neurotropic effects of this protein[33–36]. FKBP51 also facilitates expression of NF-κB, although the specific mechanism and tissue specificity of this interaction remain controversial[30].

A recent analysis distinguishing transcriptomic patterns in postsurgical patients with septic shock versus shock of other etiology showed elevated expression of *IL1R2, LCN2, LTF, MMP8,* and *OLMF4* in cases of septic shock[37]. In our analyses, IL1R2, LCN2, LTF, and MMP8 were associated with all SM subgroup comparisons to uncomplicated malaria. OLMF4 was elevated in CM and concurrent CM and SMA in comparison to uncomplicated controls. In light of the reported association with septic shock, our associations of elevated IL1R2, LCN2, LTF, MMP8, and OLMF4 in SM subtypes compared to uncomplicated controls may reflect a general signature of the inflammatory reaction to severe infection and may not represent a malaria-specific pathway.

The most significant transcript uniquely associated with cerebral malaria was IL18R1. IL18 receptors have previously been identified in murine cortical and thalamic neurons, and along with IL18 have been shown to play a role in worsening hypoxic-ischemic brain injury, one of the purported pathophysiological processes in cerebral malaria[38]. X-linked inhibitor of apoptosis-associated factor 1 (XAF1) was the most significant transcript uniquely associated with protection from concurrent cerebral malaria and severe malarial anemia. XAF1 inhibits caspase-3, -7, and -9 activity, and has been associated with neuroprotection after hypoxic-ischemic injury in mice[39]. Thrombospondin type 1 domain-containing protein (THSD4) was the most significant transcript uniquely associated with severe malarial anemia. Polymorphisms in *THSD4* have been previously associated with pulmonary function and disease[40,41]. Functionally, the protein promotes the assembly of microfibrils and has not been previously associated with malaria[42].

Pathway analyses of transcriptomic results demonstrated enrichment of processes involved in both inflammation and dampening the response to sepsis. GO analyses comparing gene expression in CM cases to uncomplicated malaria revealed that T cell activation, positive regulation of cytokine production, along with negative regulation of immune system processes were among the top-associated pathways. Uppermost associated pathways in SMA and concurrent CM and SMA also feature pro- and anti-inflammatory processes, suggesting that immune dysregulation associates with severe disease. GO analyses on protein levels revealed upregulation of the response to hypoxia and increased oxygen demand in CM cases, with no statistically significant difference in hemoglobin levels between cases and controls. This result reinforces the likelihood of ischemia-driven pathogenesis of some CM cases. Inflammatory responses, tissue hypoxia, and increased oxygen demand increase cellular energy requirements, inducing both protein metabolism and biosynthesis[43]. Pathway analysis of metabolomic data revealed that increased biosynthesis of aromatic amino acids and metabolism of alanine, aspartate, and glutamate were associated with CM.

Unpaired comparison of CM to SMA revealed that transcripts of *SLC2A1* were expressed at significantly higher levels in SMA. *SLC2A1* encodes for glucose transporter 1, GLUT1, a protein expressed in high concentrations on erythrocyte membranes, blood-brain barrier endothelial cells, and astrocytes[44–46]. GLUT1 is important in glucose transport for parasitized erythrocytes, as in vitro and murine models of *P. berghei* malaria demonstrated that inhibition of GLUT1 decreased erythrocyte glucose uptake, increased redox species, and induced erythrocyte apoptosis[47]. GLUT1 regulates gradient-dependent D-glucose delivery to the brain[46,48], and its deficiency is associated with pediatric-onset epileptic encephalopathies and exertion-induced dyskinesia[49,50]. Given that seizures are the most common clinical manifestation of neonatal hypoglycemia, the association of lower *SLC2A1* transcript levels in CM in comparison to SMA suggests that hypoglycorrhachia may underlie some of the seizures and loss of consciousness seen in CM[51].

Transcripts of genes encoding neutrophil granule protein *MPO* and *ELANE* were expressed at higher levels in concurrent CM and SMA in comparison to uncomplicated malaria, and separately in comparison to cases of CM alone. *ELANE* encodes for neutrophil elastase, a protein previously associated with retinopathy-positive cerebral malaria and undifferentiated severe malaria in comparison to unmatched controls with mild disease[7,16]. *MPO* encodes for myeloperoxidase, serum levels of which have been previously associated with cerebral malaria[52]. To our knowledge, these granule proteins have not previously been associated with the concurrent CM and SMA phenotype. The observed differences in expression between concurrent CM and SMA versus CM could be a marker for different

times between initial infection and presentation, stem from underlying variation in the genes, and/or underscore a pathophysiological difference between them.

CD14 was the only statistically significant association with cerebral malaria in both proteomic and transcriptomic analyses. CD14 is a protein expressed on the surface of macrophages and other mature cells of monocyte lineage[53]. It has been previously shown to play a role in the recognition and clearance of apoptotic cells without inducing an inflammatory cytokine response from macrophages[54]. Along with TLR4, CD14 is part of the receptor complex for gram-negative bacterial lipopolysaccharide (LPS), and in this context, it has been shown to stimulate macrophage-induced inflammation. CD14 knock-out mice have significantly lower rates of *Plasmodium berghei* cerebral malaria and parasitemia[55]. Pretreatment with LPS in experimental murine models of *P. berghei* malaria protects against cerebral malaria[56].

Proteomic analyses recapitulated prior associations of elevated serum levels of ICAM-1 and VCAM-1 with cerebral malaria[57–61]. ICAM-1 is a host endothelial receptor for *P. falciparum* erythrocyte membrane protein variants that plays a role in parasite cytoadherence to brain microvasculature[62,63]. In animal models of ischemic cerebrovascular disease, ICAM-1 has been implicated in arterial vasospasm[64]. VCAM-1 mediates adhesion of immune cells to endothelial cells. The *VCAM-1* promoter has two NF-κB binding sites, and this binding is essential for cytokine-induced transcription of VCAM-1[65]. We reported the transcriptomic association of CM with elevated FKBP51 levels, a tissue-specific NF-κB inducer, suggesting a possible signaling pathway. VCAM-1 levels have been previously associated with CM severity[60,61].

Metabolomic analysis demonstrated significantly decreased levels of L-arginine in CM compared to uncomplicated malaria controls, which supports the previously established association between low serum L-arginine levels and CM[66,67]. L-arginine is a substrate for nitric oxide production through the action of nitric oxide synthase, and L-arginine depletion, and subsequent inability to produce nitric oxide, likely leads to impaired vasodilation, which is part of the pathogenic cascade observed in CM[68]. Arginase-1, encoded by *ARG1*, catalyzes the hydrolysis of L-arginine to L-ornithine and urea as part of the urea cycle. In our analyses, *ARG1* was expressed at higher levels in all subtypes of SM, including the corresponding transcriptomic comparison of CM to uncomplicated controls without a history of CM. Previous studies have suggested that low serum arginine results from limited bioavailability of glutamine and proline, which are used to synthesize arginine[69]. Our findings suggest that L-arginine depletion may also result from increased levels of arginase-1, which is a putative mechanism for impaired cerebral vasodilation observed in CM. Furthermore, rat models of acute pulmonary embolism demonstrated that L-arginine attenuates MMP9 activity through increased nitric oxide synthesis[70]. Cell culture experiments have also demonstrated that low nitric oxide concentrations lead to increased enzymatic activity of MMP9 and suppression of exogenous TIMP1, while high nitric oxide levels attenuate MMP9 activity[71].

Elevated serum cortisol levels have been previously associated with comparisons of cerebral malaria and uncomplicated malaria to healthy controls[72]. Here we report a statistically significant difference between CM and uncomplicated controls, with higher cortisol levels in CM. Cortisol levels are known to have a circadian variation and are elevated in comatose patients[73]. The role of adrenal hormones in conferring tolerance to malaria has been established in murine models, where the murine equivalent of cortisol maintains normoglycemia and dampens cerebral inflammation[74]. However, randomized clinical trials using adjunctive corticosteroid therapy in cerebral malaria failed to show any benefit, reporting an increased risk of gastrointestinal bleeding, pneumonia, and prolonged coma[75,76].

Pipecolic acid was elevated in the sera of CM cases compared to uncomplicated malaria patients without a history of CM. This recapitulates the association of serum pipecolic acid levels established in CM patients in Malawi, as well as a relationship between increasing brain pipecolic acid levels and diminished consciousness in a murine model of malaria[77]. Pipecolic acid is likely a byproduct of hemoglobin degradation in a *P. falciparum* infected erythrocyte[78,79].

The strengths of this study include a combined analysis of transcriptomic, proteomic, and metabolomic data in a large number of participants, with 22 of 28 overlapping for all three sets of CM analyses. This allowed us to identify new associations and recapitulate prior findings, with the caveat that we can now better understand the interplay between concurrent processes in a cross-sectional fashion. The strengths of the transcriptomic analyses include the matched design, large sample sizes, and the ability to assess three different SM phenotypes.

The optimal approach to studying CM pathophysiology would involve blood and brain tissue analyses. This could be carried out in animal models and is a possible future direction for this study, although current animal models lack the parasite variant surface antigens unique to *P. falciparum* that likely mediate SM. Another area for future follow-up stems from our use of untargeted protein and metabolome analyses. While our approach was more tailored for hypothesis generation, it limited our analysis in that we had transcriptome data for only six of the 180 proteins assessed in the proteomic analyses, and we did not have protein data for the top-associating transcripts. Replication of the metabolite and protein associations with CM via targeted approaches is essential toward producing generalizable results and will be included in planned follow-up studies of severe malaria. Of note, our transcriptomic findings replicate published associations of MMP8 and MMP9 protein with CM, as well as an established association of TIMP1 transcript levels in an experimental murine cerebral malaria[14,15,80]. Other future directions stemming from this work involve additional -omics studies of severe malaria across sites in sub-Saharan Africa and Southeast Asia to validate the newly associated transcripts, proteins, and metabolites across different populations.

## Methods

### Study participants

Written informed consent was obtained from the parent or guardian of each study participant. All study protocols have been approved by the Institutional Review Board at the University of Maryland and the Faculty of Medicine, Pharmacy and Dentistry, in Bamako, Mali.

The study participants were children aged 6 months to 5 years old selected from a case-control study of cerebral malaria in Mali, enrolled from 2015 to 2018[81]. Cases of severe malaria were classified using modified World Health Organization criteria[1]. Cerebral malaria was defined by the presence of parasitemia, a Blantyre Coma Scale ≤2, and malarial retinopathy on fundoscopic examination in the absence of other obvious causes of coma. Severe malarial anemia was defined by the presence of parasitemia and a hemoglobin ≤5 g/dL. Cases of concurrent cerebral malaria and severe malarial anemia met the inclusion criteria for both severe malaria subtypes. Uncomplicated malaria was defined by parasitemia, an axillary temperature of 37.5 °C, and/or symptoms leading to seeking treatment[1]. Controls without a history of CM were defined as uncomplicated malaria without a confirmed prior episode of cerebral malaria, as ascertained by questionnaire validated for this purpose[82]. Controls with a history of CM were defined as uncomplicated malaria with a confirmed prior episode of cerebral malaria, as confirmed by questionnaire. As part of the case-control study, clinicians matched cases of severe malaria to uncomplicated malaria controls, either with or without a history of CM, based on age, sex, and ethnicity. Sex was reported by parents/guardians. We also collected data on parasitemia, blood type, hemoglobin level, and the sample collection site. Cases of severe malaria with respiratory distress were excluded from this study because we could not rule out pneumonia at enrollment.

## RNA extraction and sequencing

A total of 2.5 mL of venous blood was collected from each study participant during acute illness and before initiation of antimalarial treatment. Blood was collected into PreAnalytiX PAXgene Blood RNA collection tubes (Qiagen, QGN-762164). RNA extractions were performed with the PreAnalytiX PAXgene Blood RNA extraction kit (BD 762165) in accordance with the manufacturer's instructions. Sample quality was assessed with the RNA integrity number (RIN) obtained using an Agilent Bioanalyzer. The TruSeq Stranded mRNA Library Prep Kit (Illumina, 20020595) was used for cDNA synthesis and library preparation. Sequencing was performed using the NovaSeq6000 for 80 samples and Illumina HiSeq4000 for 4 samples at the Institute for Genome Sciences at the University of Maryland School of Medicine. Reads were mapped to the human genome Hg38 using HISAT2(v2.0.4), and counts were obtained using the Python library HTseq[83].

## Differential gene expression analyses

Samples with a read count of <50,000 were removed. In paired analyses, matched samples were removed if either the case or control had a read count of <50,000. There were 26,998 transcripts assayed for each participant. Library sizes were quality controlled and normalized using the trimmed mean of M-values method in the Bioconductor package edgeR (Supplemental Table 35)[84]. Differential gene expression analyses on transcript copy numbers were performed in edgeR using quasi-likelihood F-tests, adjusting for cell type fractions. Statistical significance was assessed using a false discovery rate threshold of 0.05 to adjust for multiple testing, as has been done in other malaria expression analyses[7,85]. Gene function was determined using OMIM[42]. Pathway analyses were performed using the Biological Process domain of Gene Ontology and the KEGG database, with results adjusted using a false discovery rate $p$-value threshold[86,87].

## Proteomics in cerebral malaria

Serum protein levels were assessed with non-targeted quantitative proteomic analysis using data-independent acquisition with liquid chromatography with tandem mass spectrometry (LC-MS/MS) on an Orbirap mass spectrometer. All samples were analyzed in triplicate.

**In solution digestion.** The protein concentration in each serum sample was estimated to be 70ug/ul for protein content. A volume corresponding to 100 μg of each sample was removed. The disulfides were reduced in 100 mM dithiothreitol (Sigma, 43815-5G) for 30 min at 37 °C. The free sulphydryls were then alkylated with 240 mM iodoacetamide (Fluka, 57670) for 30 min at room temperature in the dark. The proteins were then digested with Trypsin (Promega, V5111) at a protein:enzyme ratio of 10:1 overnight at 37 °C. 10% formic acid (Fisher, A117-50) was then added to halt the digestion. The peptides were then cleaned using a Water Oasis HLB plate and eluted in 60% acetonitrile/0.1% formic acid (Fisher, A955-4 and Fisher, A117-50, respectively, dilutions made using sterile water, Fisher, W6-4) and brought to dryness on a speedvac. The peptides were then resuspended at a concentration of 1 μg/μl in 0.1% formic acid. A DIA Chromatogram Library sample was created from a pooled aliquot of the samples.

**LC-MS/MS analysis, Orbitrap fusion.** A 1 μL injection was separated by online reverse phase chromatography using a Thermo Scientific EASY-nLC 1000 system with a reversed-phase pre-column Magic C18-AQ (100 μm I.D., 2.5 cm length, 5 μm, 100 Å), and an in-house prepared reverse-phase nano-analytical column Magic C-18AQ (75 μm I.D., 20 cm length, 5 μm, 100 Å, Michrom BioResources Inc, Auburn, CA), at a flow rate of 300 nl/min. The chromatography system was coupled online with an Orbitrap Fusion Tribrid mass spectrometer (Thermo Fisher Scientific, San Jose, CA) equipped with a Nanospray Flex NG source (Thermo Fisher Scientific). Solvents were A: 2% Acetonitrile, 0.1%

Formic acid; B: 90% Acetonitrile, 0.1% Formic acid. Samples were separated by a 64-min gradient (0 min: 5% B; 50 min: 25% B; 52 min: 40% B; 54 min: 90% B; hold 5 min: 90% B; 1 min: 5% B; 4 min: 5% B). Data was acquired using a data-independent acquisition (DIA) strategy. The Orbitrap Fusion instrument parameters (Fusion Tune 3.3 software) were as follows for Orbitrap (OT-MS) Oontrap (OT- MS/MS) with HCD fragmentation: Nano-electrospray ion source with spray voltage 2.55 kV, capillary temperature 275. To create a chromatogram library, the Orbitrap Fusion Tribrid was configured to acquire six chromatogram library acquisitions with a 60,000 resolution full MS1spectrum matching the range (i.e., 395–505,495–605, 595–705, 695–805, 795–905, and 895–1005 m/z) using an AGC target of 4e5 and a maximum inject time of 60 ms. The MS2 acquisition strategy for DIA spectra (4 m/z precursor isolation windows at 30,000 resolution, AGC target 4e5, maximum inject time 60 ms, Charge state 3, HCD 33) using an overlapping window pattern from narrow mass ranges using window placements optimized by EncyclopeDIA (i.e.,396.43–502.48, 496.48–602.52, 596.52–702.57, 696.57–802.61, 796.61–902.66, and 896.6–1002.70 m/z)[88]. For quantitative samples, the Orbitrap Fusion Tribrid was configured to acquire a 60,000 resolution full MS1spectrum 385-1015mz range, AGC target 4e5, maximum inject time 60 ms) and matching the range 25 × 24 m/z DIA spectra (24 m/z precursor isolation windows at 30,000 resolution, AGC target 4e5, maximum inject time 60 ms, Charge state 3, HCD 33) using an overlapping window pattern from 388.43 to 1012.70 m/z using window placements optimized by EncyclopeDIA.

**Data analysis parameters.** A human FASTA database was downloaded from Uniprot (http://uniprot.org). This file was used with the 6 gas phase fraction files from the analysis of the chromatogram library sample to create a human plasma-specific chromatogram library using the DIA-NN (v 1.8) software package[89]. This chromatogram library file was then used to perform identification and quantitation of the proteins in the 108 samples again, using DIA-NN as the acquisition type, trypsin as the enzyme, CID/HCD as the fragmentation, 10 ppm mass tolerances for the precursor, fragment, and library mass tolerances. The precursor FDR rate was set to 1%.

**Data quality control.** The DIA-NN protein output table was imported into in-house built software to enable visualization of the data to assess overall data quality and to check for outliers.

**Statistical analysis.** Proteins with ambiguous assignments were removed from the analysis. Triplicate values were averaged, and the log2 of the fold change is presented below. Statistical significance was assessed with a two-tailed paired $t$-test, using an alpha level of 0.05. Gene ontology analyses were performed using topGO and GOstats packages, and KEGG pathway analysis was performed using gage, gagedata, and pathview packages in R. Detailed methods are available in the Supplementary Methods.

## Metabolomics in cerebral malaria

**Serum metabolite levels were assessed with a non-targeted quantitative approach**

**Sample preparation.** Human serum samples were taken out of a −80 °C freezer and placed on ice. 100 μL was aliquoted to a 1.5-mL Eppendorf safe-lock tube. 900 μL of mixed methanol/chloroform (6:2) was added (Fisher, A456-4 and Sigma, 366927-1L, respectively). The tube was vortex mixed for 20 s at 3000 rpm, sonicated in an ice-water bath for 3 min, and then centrifuged at 21,000 × g and 10 °C for 15 min. A 300-μL aliquot of the clear supernatant was precisely taken out and transferred to an LC injection micro-vial and dried down under a gentle nitrogen gas flow in a nitrogen evaporator at 30 °C. The residue was dissolved in 60 μL of methanol. 6 μL was injected to run reversed-phase LC-MS for detection and relative quantitation of polar to

lipophilic metabolites. A 500-µL aliquot of the clear supernatant was precisely taken out and transferred to another 1.5-mL Eppendorf safe-lock tube. 250 µL of water (Fisher, W6-4) and 200 µL of chloroform were added to the tube, and the mixture was vortex mixed for 20 s followed by centrifugal clarification at 21,000 × g and 10 °C for 6 min to split the whole phase into two phases. The upper aqueous phase was carefully taken out to an LC injection micro-vial and dried down in a nitrogen evaporator at 30 °C. The residue was reconstituted in 100 µL of 50% acetonitrile. 5 µL was injected to run HILIC-MS for detection and relative quantitation of very polar and hydrophilic metabolites.

**LC-MS system.** A Dionex Ultimate 3000 UHPLC system coupled to a Thermo Scientific LTQ-Orbitrap Velos Pro mass spectrometer equipped with an electrospray ionization (ESI) source was used.

**RPLC-MS of polar to lipophilic metabolites.** Reversed-phase (RP)-UPLC-MS runs were carried out for analysis of polar to hydrophobic metabolites with the use of a Waters C8 UPLC column (2.1 × 50 mm, 1.7 µm) for chromatographic separation. The mobile phase was (A) 0.1% formic acid in water (Fisher, A117-50) and (B) 0.1% formic acid in acetonitrile-isopropanol (1:1, v/v, Fisher, A955-4 and Fisher, A461-4, respectively) for positive-ion detection. For negative-ion detection, the mobile phase was (A) 0.01% formic acid in water and (B) 0.01% formic acid in acetonitrile-isopropanol (1:1, v/v). The efficient gradient was 5% to 50% B in 5 min; 50% to 100% B in 15 min, and 100% B for 2.5 min before the column was equilibrated for 4 min at 5% B between injections. The column flow rate was 400 µL/min, and the column temperature was maintained at 45 °C. The LTQ Velo Pro Orbitrap MS instrument was operated in the survey-scan mode with full-mass and high-resolution Fourier transform (FT) MS detection at a mass resolution of 60,000 FWHM @ m/z 200. The mass scan range was 80–1800 m/z for positive-ion and 70–1000 m/z negative-ion detection. Along with the LC-MS data acquisitions, LC-MS/MS data were also acquired for several samples using collision-induced dissociation (CID). For LC-MS/MS, the 6 most abundant ions from each survey scan were chosen for subsequent CID in each duty cycle, with the normalized collision energies of 28%–35%.

**LC-MS of very polar metabolites.** For analysis of very polar metabolites, Hydrophilic interaction chromatography (HILIC) –MS using a Waters Amide column (2.1 × 100 mm, 1.7 µm) was performed, with positive-ion or negative-ion detection in each round of two LC injections per sample. For HILIC-MS, the mobile phase was (A) 0.01% formic acid in water and (B) 0.01% formic acid in acetonitrile. The efficient gradient was 90% for 1 min; 90% to 20% B in 9 min; 20% B for 2.5 min before the column was equilibrated at 90% B for 6 min between injections. The column flow rate was 330 µL/min, and the column temperature was maintained at 45 °C. The MS instrument was operated in the survey-scan mode with full-mass FTMS detection at a mass resolution of 60,000 FWHM @ m/z 200. The mass scan range was m/z 70–1000 for both positive-ion and negative-ion detection. Along with the LC-MS data acquisitions, LC-MS/MS data were also acquired for a few samples using collision-induced dissociation (CID). For LC-MS/MS runs, the 6 most abundant ions from each survey scan were chosen for subsequent CID in each duty cycle, with the normalized collision energies at 25%–30%.

**Statistical analysis.** Metabolites with ambiguous name assignments were removed. Statistical significance was assessed with a paired, two-tailed t-test, and false discovery rate was used for multiple testing adjustment using an alpha level of 0.05. Log base 2 fold change is reported throughout. Enrichment and pathway analyses were performed using MetaboAnalyst 5.0. Enrichment analyses interrogated the chemical structure sub-class category and pathway analyses were performed using the KEGG *Homo sapiens* pathway library and a hypergeometric test[87].

### Power analysis
Comparisons of gene expression between cases and matched uncomplicated malaria controls had 80% power to detect a significant mean difference that ranged from 24% for cerebral malaria ($N = 14$ pairs; two-sided $p < 0.05$; standard deviation = 0.3) to 27% for severe malarial anemia ($N = 12$ pairs) to 35% for concurrent CMA and SMA infection ($N = 8$ pairs).

Unpaired comparisons of gene expression between severe malaria groups had 80% power to detect a significant mean difference of 27% for cerebral malaria versus severe malarial anemia (21 CM cases vs 20 SMA cases; two-sided $p < 0.05$; standard deviation = 0.3); 33% for cerebral malaria versus concurrent disease (21 CM cases vs 10 concurrent CM and SMA cases); and 34% for severe malarial anemia versus concurrent disease (20 SMA cases vs 10 concurrent CM and SMA cases).

Published power analyses of unpaired MS/MS data using the MultiPower R package demonstrated that the power estimated for an unpaired metabolomics experiment given a sample size of 14 was >0.95, and estimated an optimal sample size for unpaired proteomic experiments to be 16[90]. The statistical power of our approach to detect a difference in variables between study groups is further strengthened by our use of a matched design.

### Deconvolution analysis
B cell, T cell, monocyte, and neutrophil cell fractions were estimated using CibersortX[91]. The hematopoietic gene signature matrix LM22 was used as a signature matrix. Sample distributions before and after adjustment for deconvolution were compared with principal component analyses, using the ggplot2 package in R. Mean proportions of T cells, B cells, and neutrophils were associated with each severe malaria subtype using the t-test.

### Reporting summary
Further information on research design is available in the Nature Portfolio Reporting Summary linked to this article.

## Data availability
RNA-Seq data is available in the Gene Expression Omnibus, proteomic data is available in PRIDE, and metabolomic data is available MassIVE. Metabolomics Data. Data Repository: MassIVE. Dataset ID: MSV000096913. url: https://massive.ucsd.edu/ProteoSAFe/dataset.jsp?task=ba8df9f0096c4bf6abb946f323f153c1. Made public: 03/25/2025. Proteomics Data. Data repository: Pride. Dataset ID: PXD058621. url: https://www.ebi.ac.uk/pride/archive/projects/PXD058621. Project accession: PXD058621. Release Date: 03/25/2025. Transcriptomic data: Data repository: GEO. Dataset ID: GSE289197. Reviewer access details: GSE289197. url: https://www.ncbi.nlm.nih.gov/geo/query/acc.cgi?acc=GSE289197. Release Date: 03/25/2025 Source data are provided with this paper.

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

## Acknowledgements

We thank the team of the severe malaria case-control study in Mali and the UMSOM CVD Malaria Research Program. We also thank Ana Raquel da Costa, Melissa Myers, Tina Williams, Joanne Morrison, and Nicole Eddington Johnson for administrative support. This work was supported by National Institutes of Health grants R01HL130750 (C.V.P.; NHLBI) and R01HL146377 (M.A.Tr.; NHLBI); cooperative agreement U19AI065683 (C.V.P.; NIH: NIAID); NIH IRID grant R01AI099628 (M.A.Th.); a Burroughs Wellcome Fund/American Society of Tropical Medicine and Hygiene Postdoctoral Fellowship to M.A.Tr.; and an award to C.V.P. from the Howard Hughes Medical Institute. This research was supported by a subproject to R.S.S. on NIH R25NS070695 and by a postdoctoral fellowship awarded to E.M.S. through the NIH/NIAID T32AI007524 Fellowship Training Program in Vaccinology. This publication was made possible by the University of Maryland Baltimore Institute for Clinical and Translational Research (ICTR), which is funded in part by Grant Number TL1 TR003100 (M.A.Tr.) from the National Center for Advancing Translational Sciences (NIH/NCATS).

## Author contributions

Conceptualization: R.S.S., D.C., E.M.S., M.A.Th., M.A.Tr. Methodology: R.S.S., D.C., E.M.S., J.G.L., B.E.C., S.S., A.D., J.B.M., A.O., A.K.K., B.K., K.T., B.G., B.M.T., A.N., N.T.V., M.D., I.D., Y.T., M.S., F.M., A.D., A.Tr., A.Th., M.B.L., K.E.L., B.K., O.K.D., C.V.P., D.R.G., J.C.S., M.A.Th., M.A.Tr. Investigation: A.D., J.B.M., A.O., A.K.K., K.T., B.G., M.B.T., A.N., M.D., I.D., Y.T., M.S., M.B.L., B.K., O.K.D., J.C.S., M.A.Th. Visualization: R.S.S., M.A.Tr. Funding acquisition: R.S.S., M.D., M.S., O.K.D., C.V.P., M.A.Th., M.A.Tr. Project administration: D.C., A.D., A.O., A.K.K., K.T., B.G., B.M.T., A.N., N.T.V., M.D., I.D., Y.T., M.S., M.B.L., B.K., O.K.D., D.R.G., M.A.Th., M.A.Tr. Supervision: M.D., M.S., O.K.D., C.V.P., M.A.Th., M.A.Tr. Writing original draft: R.S.S., M.A.Tr. Writing – review & editing: R.S.S., D.C., E.M.S., J.G.L., B.E.C., S.S., A.D., J.B.M., A.O., A.K.K., B.K., K.T., B.G., B.M.T., A.N., M.D., I.D., Y.T., M.S., F.M., A.D., A.Tr., A.Th., M.B.L., K.E.L., B.K., O.K.D., C.V.P., D.R.G., J.C.S., M.A.Th., M.A.Tr.

## Competing interests

The authors declare no competing interests.

## Additional information

¹Malaria Research Program, Center for Vaccine Development and Global Health, University of Maryland School of Medicine, Baltimore, MD, USA. ²Ken and Ruth Davee Department of Neurology, Northwestern University, Chicago, IL, USA. ³Malaria Research and Training Center, University of Sciences Techniques and Technologies, Bamako, Mali. ⁴Institute for Genome Sciences, University of Maryland School of Medicine, Baltimore, MD, USA. ⁵Department of Biochemistry and Microbiology, University of Victoria, Victoria, BC, Canada. ⁶These authors jointly supervised this work: Mahamadou A. Thera, Mark A. Travassos. ⁷Deceased: Modibo Daou. ⁸Deceased: Mody Sissoko. ⁹Deceased: Ogobara K. Doumbo. ✉e-mail: mtravass@som.umaryland.edu

