## [Transparent Peer Review file · Nature Communications]

A shared inflammatory signature across severe malaria syndromes in transcriptomic, proteomic, and metabolomic analyses

Corresponding Author: Dr Mark Travassos

Version 0:

Reviewer comments:

Reviewer #2

(Remarks to the Author)

The manuscript presents a transcriptomic, proteomic, and metabolomic analysis of different subgroups of severe malaria matched with controls in children from Mali. The authors describe the molecular differences between the groups and discuss potential functions in malaria syndromes.

Although the data generated is interesting to potentially identify biological markers, and address future biological questions, it is difficult to assess for example the quality of the untargeted metabolomics since there are no details on how the samples were collected, processed, and how these methods are different or similar between groups. Many different factors can affect metabolomics and those are not explained in the manuscript.

The authors did not include a sample size calculation or an explanation of why this is lacking, on how this may affect conclusions in the manuscript.

There is no attempt to validate at least, some of the results, or discussion on the need of replication studies or generalizability of the observed results.

Reviewer #3

(Remarks to the Author)

Sobota et al. present transcriptomic, proteomic, and metabolomic analyses comparing subtypes of severe *Plasmodium falciparum* malaria to matched controls with uncomplicated cases. The manuscript presents interesting findings, and I have the following suggestions that may be used to improve it.

Introduction can be strength by explaining the limitations of existing data.

Authors may want to include a line on different confounding factors used for p-value adjustment, if any.

Based on the number of samples included in the statistical analysis to identify significant differences in transcriptomic, proteomic, and metabolomic data, the power of the study may be provided separately for each transcriptomic, proteomic, and metabolomic. This will help readers comprehend the significance of the results.

If the calculated statistical power of the study is low, I believe that conducting a study to validate the identified signatures on additional cohorts would be necessary to strengthen the significance of this research.

Reviewer #4

(Remarks to the Author)

This manuscript reports transcriptional(whole blood), proteomic, and metabolomic differences between different clinical subtypes of severe *Plasmodium falciparum* malaria to matched controls with uncomplicated disease in 79 children from Mali to inform disease models.

For CM analysis 12 matched pairs were analyzed (405 differential genes). concurrent CM and SMA to uncomplicated controls, no differential genes identified; 8 cases of severe malaria anemia to controls compared with 19 differentially abundant genes identified; 21 cases of CM to 20 cases of SMA were compared; 21 cases of CM and 10 cases of concurrent CM and SMA (19 differential genes). 16 pairs compared with history of CM compared to no history of CM.

Major

- Writing could be improved, presently it is a list of data for each clinical subtype vs control or vs another clinical subtype, difficult to determine what the main messages are, and most data is transcriptional
- Interesting comparison of clinical phenotypes in malaria using 3 modalities (RNA, Protein Metabolites). They list genes, proteins and metabolites for each comparison; Could be strengthened by integrating the analysis (ie pathways and metabolite data, to leverage that all this data is taken into account in an integrated way.
- I think the novel data that they want to highlight is the role of metalloproteinases in the CNS, can they expand on this, what cell type is involved, how does this peripheral signature reflect CNS pathology-if this is not the main message, what is and can it be supported with some of the different datasets?

Can they integrate the metabolomics data with the transcriptional data (particularly where they identify differential metabolite pathways). Ie a main finding from the abstract L-arginine metabolites were decreased in cerebral malaria, which coupled with increased ARG1 transcription suggests a putative mechanism impairing cerebral vasodilation. (ie they highlight a main finding in the abstract

they did not have protein data for the top associating transcripts from their analysis. If they could confirm protein differences for some of the transcripts (the ones that may inform their new disease a model information for example) (ie they highlight a main finding in the abstract "Tissue inhibitor of metalloproteinases 1 was the most upregulated protein in cerebral malaria, which along with elevated MMP8 and MMP9 transcription, underscores the importance of the metalloproteinase pathway" can any of these proteins differentials be confirmed?

Table 1, as ethnicity is not compared between patients and most are Dogon, no need to specify their ethnicity; same with blood type, unless the comparison between blood type was done, or would have a major effect on one of the results. Same with Rh factor.

Version 1:

Reviewer comments:

Reviewer #2

(Remarks to the Author)

The authors have addressed all my comments satisfactorily.

Reviewer #3

(Remarks to the Author)

No further comments

Reviewer #4

(Remarks to the Author)

REVIEWER COMMENTS

Reviewer #2 (Remarks to the Author):

The manuscript presents a transcriptomic, proteomic, and metabolomic analysis of different subgroups of severe malaria matched with controls in children from Mali. The authors describe the molecular differences between the groups and discuss potential functions in malaria syndromes.

Although the data generated is interesting to potentially identify biological markers, and address future biological questions, it is difficult to assess for example the quality of the untargeted metabolomics since there are no details on how the samples were collected, processed, and how these methods are different or similar between groups. Many different factors can affect metabolomics and those are not explained in the manuscript.

We appreciate the reviewer's comment and have added the following to the Supplemental Methods section to more fully explain the data generation and processing in the proteomic and metabolomic experiments:

"Proteomics

In solution digestion:

The protein concentration in each serum sample was estimated to be 70ug/ul for protein content. A volume corresponding to 100 ug of each sample was removed. The disulphides were reduced in 100mM dithiothreitol for 30 minutes at 37°C. The free sulphhydryls were then alkylated with 240mM iodoacetamide for 30 minutes at room temperature in the dark. The proteins were then digested with Trypsin (Promega) at a protein:enzyme ratio of 10:1 overnight at 37°C. 10% formic acid was then added to halt the digestion. The peptides were then cleaned using a Water Oasis HLB plate and eluted in 60% acetonitrile/0.1% formic acid and brought to dryness on a speedvac. The peptides were then resuspended at a concentration of 1ug/ul in 0.1% formic acid. A DIA Chromatogram Library sample was created from a pooled aliquot of the samples.

LC-MS/MS analysis, Orbitrap Fusion:

A 1µL injection was separated by on-line reverse phase chromatography using a Thermo Scientific EASY-nLC 1000 system with a reversed-phase pre-column Magic C18-AQ (100µm I.D., 2.5 cm length, 5µm, 100Å, and an in-house prepared reverse phase nano-analytical column Magic C-18AQ (75µm I.D., 20 cm length, 5µm, 100Å, Michrom BioResources Inc, Auburn, CA), at a flow rate of 300 nl/min. The chromatography system was coupled on-line with an Orbitrap Fusion Tribrid mass spectrometer (Thermo Fisher Scientific, San Jose, CA) equipped with a Nanospray Flex NG source (Thermo Fisher Scientific). Solvents were A: 2% Acetonitrile, 0.1% Formic acid; B: 90% Acetonitrile, 0.1% Formic acid. Samples were separated by a 64-minute gradient (0 min: 5%B; 50 min: 25%B; 52 min: 40%B; 54 min: 90%B; hold 5min: 90%B; 1 min: 5%B; 4 min: 5%B). Data was acquired using data-independent acquisition (DIA) strategy. The Orbitrap Fusion instrument parameters (Fusion Tune 3.3 software) were as follows for orbitrap (OT-MS) iontrap (OT-MS/MS) with HCD fragmentation: Nano-electrospray ion source with spray voltage 2.55kV, capillary temperature 275. To create a chromatogram library, the Orbitrap Fusion Tribrid was configured to acquire six chromatogram library acquisitions with a 60,000 resolution full MS1 spectrum matching the range (i.e., 395–505, 495–605, 595–705, 695–805, 795–905, and 895–1005 m/z) using an AGC target of 4e5 and a maximum inject time of 60 ms. The MS2 acquisition strategy for DIA spectra (4 m/z precursor isolation windows at 30,000 resolution, AGC target 4e5, maximum inject time 60 ms,

Charge state 3, HCD 33) using an overlapping window pattern from narrow mass ranges using window placements optimized by EncyclopeDIA (Searle, BC et al. PMID: 30510204) (i.e., 396.43–502.48, 496.48–602.52, 596.52–702.57, 696.57–802.61, 796.61–902.66, and 896.6–1002.70 m/z). For quantitative samples, the Orbitrap Fusion Tribrid was configured to acquire a 60,000 resolution full MS1 spectrum 385-1015 m/z range, AGC target 4e5, maximum inject time 60 ms) and matching the range 25 × 24 m/z DIA spectra (24 m/z precursor isolation windows at 30,000 resolution, AGC target 4e5, maximum inject time 60 ms, Charge state 3, HCD 33) using an overlapping window pattern from 388.43 to 1012.70 m/z using window placements optimized by EncyclopeDIA.

Data Analysis Parameters:

A human FASTA database was downloaded from Uniprot (<http://uniprot.org>). This file was used with the 6 gas phase fraction files from the analysis of the chromatogram library sample to create a human plasma specific chromatogram library using the DIA-NN (v 1.8) software package (Demichev, V et al. PMID: 31768060). This chromatogram library file was then used to perform identification and quantitation of the proteins in the 108 samples again using DIA-NN as the acquisition type, trypsin used as the enzyme, CID/HCD as the fragmentation, 10 ppm mass tolerances for the precursor, fragment, and library mass tolerances. The precursor FDR rate was set to 1%.

Data Quality Control:

The DIA-NN protein output table was imported into in-house built software to enable visualization of the data to assess for overall data quality and to check for outliers.

Metabolomics

Sample preparation:

Human serum samples were taken out of a -80 °C freezer and placed on ice. 100 µL was aliquoted to a 1.5-mL Eppendorf safe-lock tube. 900 µL of mixed methanol/chloroform (6:2) was added. The tube was vortex mixed for 20 s at 3000 rpm, sonicated in an ice-water bath for 3 min and then centrifuged at 21,000 × g and 10 °C for 15 min. A 300-µL aliquot of the clear supernatant was precisely taken out and transferred to an LC injection micro-vial and dried down under a gentle nitrogen gas flow in a nitrogen evaporator at 30 °C. The residue was dissolved in 60 µL of methanol. 6 µL was injected to run reversed-phase LC-MS for detection and relative quantitation of polar to lipophilic metabolites. A 500-µL aliquot of the clear supernatant was precisely taken out and transferred to another 1.5-mL Eppendorf safe-lock tube. 250 µL of water and 200 µL of chloroform were added to the tube and the mixture was vortex mixed for 20 s followed by centrifugal clarification at 21,000 × g and 10 °C for 6 min to split the whole phase into two phases. The upper aqueous phase was carefully taken out to an LC injection micro-vial and dried down in a nitrogen evaporator at 30 °C. The residue was reconstituted in 100 µL of 50% acetonitrile. 5 µL was injected to run HILIC-MS for detection and relative quantitation of very polar and hydrophilic metabolites.

LC-MS system:

A Dionex Ultimate 3000 UHPLC system coupled to a Thermo Scientific LTQ-Orbitrap Velos Pro mass spectrometer equipped with an electrospray ionization (ESI) source was used.

RPLC-MS of polar to lipophilic metabolites:

Reversed-phase (RP)-UHPLC-MS runs were carried out for analysis of polar to hydrophobic metabolites with the use of a Waters C8 UPLC column (2.1 × 50 mm, 1.7 µm) for chromatographic separation. The

mobile phase was (A) 0.1% formic acid in water and (B) 0.1% formic acid in acetonitrile-isopropanol (1:1, v/v) for positive-ion detection. For negative-ion detection, the mobile phase was (A) 0.01% formic acid in water and (B) 0.01% formic acid in acetonitrile-isopropanol (1:1, v/v). The efficient gradient was 5% to 50% B in 5 min; 50% to 100% B in 15 min and 100% B for 2.5 min before the column was equilibrated for 4 min at 5% B between injections. The column flow rate was 400 $\mu\text{L}/\text{min}$ and the column temperature was maintained at 45 °C. The LTQ Velo Pro Orbitrap MS instrument was operated in the survey-scan mode with full-mass and high-resolution Fourier transform (FT) MS detection at a mass resolution of 60,000 FWHM @ m/z 200. The mass scan range was 80 to 1800 m/z for positive-ion and 70 to 1000 m/z negative-ion detection. Along with the LC-MS data acquisitions, LC-MS/MS data was also acquired for several samples using collision induced dissociation (CID). For LC-MS/MS, the 6 most abundant ions from each survey scan were chosen for subsequent CID in each duty cycle, with the normalized collision energies of 28% to 35%.

LC-MS of very polar metabolites:

For analysis of very polar metabolites, Hydrophilic interaction chromatography (HILIC) –MS using a Waters Amide column (2.1 x 100 mm, 1.7 μm) was performed, with positive-ion or negative-ion detection in each round of two LC injections per sample. For HILIC-MS, the mobile phase was (A) 0.01% formic acid in water and (B) 0.01% formic acid in acetonitrile. The efficient gradient was 90% for 1 min; 90% to 20% B in 9 min; 20% B for 2.5 min before the column was equilibrated at 90% B for 6 min between injections. The column flow rate was 330 $\mu\text{L}/\text{min}$ and the column temperature was maintained at 45 °C. The MS instrument was operated in the survey-scan mode with full-mass FTMS detection at a mass resolution of 60,000 FWHM @ m/z 200. The mass scan range was m/z 70 to 1000 for both positive-ion and negative-ion detection. Along with the LC-MS data acquisitions, LC-MS/MS data was also acquired for a few samples using collision induced dissociation (CID). For LC-MS/MS runs, the 6 most abundant ions from each survey scan was chosen for subsequent CID in each duty cycle, with the normalized collision energies at 25% to 30%.”

The authors did not include a sample size calculation or an explanation of why this is lacking, on how this may affect conclusions in the manuscript.

We appreciate the need for putting our results within the context of what we could detect given our sample sizes. We have added the following power calculations to the Supplemental Methods section.

Power calculations, transcriptomics:

“Comparisons of gene expression between cases and matched uncomplicated malaria controls had 80% power to detect a significant mean difference that ranged from 24% for cerebral malaria (N=14 pairs; two-sided $p < 0.05$; standard deviation=0.3) to 27% for severe malarial anemia (N=12 pairs) to 35% for concurrent CMA and SMA infection (N=8 pairs).

Unpaired comparisons of gene expression between severe malaria groups had 80% power to detect a significant mean difference of 27% for cerebral malaria versus severe malarial anemia (21 CM cases vs 20 SMA cases; two-sided $p < 0.05$; standard deviation=0.3); 33% for cerebral malaria versus concurrent disease (21 CM cases vs 10 concurrent CM and SMA cases); and 34% for severe malarial anemia versus concurrent disease (20 SMA cases vs 10 concurrent CM and SMA cases).”

Power estimation for proteomics and metabolomics:

“Published power analyses of unpaired MS/MS data using the MultiPower R package demonstrated that the power estimated for an unpaired metabolomics experiment given a sample size of 14 was >0.95, and estimated an optimal sample size for unpaired proteomic experiments to be 16. (Nature Communications PMID: 32555183) The statistical power of our approach to detect a difference in variables between study groups is further strengthened by our use of a matched design.”

There is no attempt to validate at least, some of the results, or discussion on the need of replication studies or generalizability of the observed results.

We thank the reviewer for this comment, and we have modified our discussion in the following way (Discussion lines 358-361):

“The optimal approach to studying CM pathophysiology would involve blood and brain tissue analyses. This could be carried out in animal models and is a possible future direction for this study, although current animal models lack the parasite variant surface antigens unique to *P. falciparum* that likely mediate SM. Another area for future follow-up stems from our use of untargeted protein and metabolome analyses. While our approach was more tailored for hypothesis generation, it limited our analysis in that we had transcriptome data for only six of the 180 proteins assessed in the proteomic analyses, and we did not have protein data for the top associating transcripts. **Replication of the metabolite and protein associations with CM via targeted approaches is essential toward producing generalizable results, and will be included in planned follow-up studies of severe malaria.** Other future directions stemming from this work involve additional -omics studies of severe malaria across sites in sub-Saharan Africa and Southeast Asia to validate the newly associated transcripts, proteins, and metabolites **across different populations.**”

Reviewer #3 (Remarks to the Author):

Sobota et al. present transcriptomic, proteomic, and metabolomic analyses comparing subtypes of severe Plasmodium falciparum malaria to matched controls with uncomplicated cases. The manuscript presents interesting findings, and I have the following suggestions that may be used to improve it.

Introduction can be strengthened by explaining the limitations of existing data.

We thank the reviewer for this comment and have adjusted our introduction in the following manner to indicate the limitations of existing data (Introduction lines: 55-57):

“Advances in high-throughput technology have facilitated characterization of transcriptomic, proteomic, and metabolomic signatures of biological processes that mediate the interaction between *P. falciparum* and the human host, but severe malarial subtypes in sub-Saharan Africa have not been comprehensively profiled in this manner⁶⁻¹¹. **To date, there are no published studies of severe malaria that describe transcriptomics, proteomics, and metabolomics in overlapping samples.** Here we present an integrated analysis using these approaches in children with CM, providing additional insight by contextualizing the findings of each approach within the greater biological framework. In addition, we performed transcriptomic analyses of SMA, concurrent CM and SMA, and uncomplicated malaria without a history of cerebral malaria. We hypothesized that transcriptomic, proteomic, and metabolomic profiles of children with SM subtypes differ from each other, and from children with uncomplicated disease.”

Authors may want to include a line on different confounding factors used for p-value adjustment, if any.

We thank the reviewer for this comment and have added citations and adjusted our methods in the following manner (Methods lines: 404-405):

“Statistical significance was assessed using a false discovery rate (FDR) threshold of 0.05 to adjust for multiple testing, **as has been done in other malaria expression analyses** (Lee, HJ et al. PMID: 29950443 and Guillochon, E et al. PMID: 35255125).”

Based on the number of samples included in the statistical analysis to identify significant differences in transcriptomic, proteomic, and metabolomic data, the power of the study may be provided separately for each transcriptomic, proteomic, and metabolomic. This will help readers comprehend the significance of the results. If the calculated statistical power of the study is low, I believe that conducting a study to validate the identified signatures on additional cohorts would be necessary to strengthen the significance of this research.

We appreciate the reviewer’s comments, and per our reply to the comments from Reviewer #2 above, have now included information on power calculations for each of the comparisons in the manuscript. As indicated above, the matched design of the study allowed detection of relatively subtle differences between groups.

Reviewer #4

This manuscript reports transcriptional(whole blood), proteomic, and metabolomic differences between different clinical subtypes of severe Plasmodium falciparum malaria to matched controls with uncomplicated disease in 79 children from Mali to inform disease models.

For CM analysis 12 matched pairs were analyzed (405 differential genes). concurrent CM and SMA to uncomplicated controls, no differential genes identified; 8 cases of severe malaria anemia to controls compared with 19 differentially abundant genes identified; 21 cases of CM to 20 cases of SMA were compared; 21 cases of CM and 10 cases of concurrent CM and SMA (19 differential genes). 16 pairs compared with history of CM compared to no history of CM.

Major

• Writing could be improved, presently it is a list of data for each clinical subtype vs control or vs another clinical subtype, difficult to determine what the main messages are, and most data is transcriptional

We appreciate the reviewer's comment and have focused on presenting the main messages in the Discussion, synthesizing the findings presented in the Results section. Given the many facets of the Results section, we felt that this was the most effective way to present the study. We have reviewed the manuscript and have attempted to clarify the Results and Discussion language as best we can.

• Interesting comparison of clinical phenotypes in malaria using 3 modalities (RNA, Protein Metabolites). They list genes, proteins and metabolites for each comparison; Could be strengthened by integrating the analysis (ie pathways and metabolite data, to leverage that all this data is taken into account in an integrated way).

We thank the reviewer for this comment and have added the following metabolite pathway analyses into the manuscript (Results Line 201), along with a discussion on how they integrate with pathway analyses from transcriptomic and proteomic analyses (Discussion Line 276), and Supplemental Table S11 in the Supplemental Materials file:

Results:

Metabolomic profiles were ascertained for the same set of 14 CM cases and 14 age-, sex-, and ethnicity-matched uncomplicated malaria controls without a history of CM as in the proteomic analysis. After removing ambiguous assignments, 147 lipids and metabolites met the multiple testing-corrected level of significance. Nervonic acid (logFC 0.89, p-value 0.026), pipercolic acid (logFC 1.76, p-value 0.044), cortisol (logFC 1.59, p-value 3.61E-03), and mannitol (logFC 7.43, p-value 0.014) were among the metabolites with significantly higher levels in CM (Supplemental Table S10A). Paracetamol was also higher in CM cases (logFC 7.37, p-value 0.035). Arginine and linoleic acid were among the metabolites with lower

levels in cases of CM compared to uncomplicated controls (logFC -1.04 and -0.79, p-value 0.038 and 0.017, respectively; Supplemental Table S10B). **Pathway analysis of serum metabolites in CM compared to uncomplicated malaria controls revealed that phenylalanine, tyrosine, and tryptophan biosynthesis, and alanine, aspartate, and glutamate metabolism were among significantly upregulated pathways in cerebral disease (Supplemental Table S11).**

Discussion:

Pathway analyses of transcriptomic results demonstrated enrichment of processes involved in both inflammation and dampening the response to sepsis. GO analyses comparing gene expression in CM cases to uncomplicated malaria revealed that T cell activation, positive regulation of cytokine production, along with negative regulation of immune system process were among the top associating pathways. Uppermost associated pathways in SMA and concurrent CM and SMA also feature pro- and anti-inflammatory processes, suggesting that immune dysregulation associates with severe disease. GO analyses on protein levels revealed upregulation of the response to hypoxia and increased oxygen demand in CM cases, with no statistically significant difference in hemoglobin levels between cases and controls. This result reinforces the likelihood of ischemia-driven pathogenesis of some CM cases. **Inflammatory responses, tissue hypoxia and increased oxygen demand increase cellular energy requirements, inducing both protein metabolism and biosynthesis (PMID: 31036854). Pathway analysis of metabolomic data revealed that increased biosynthesis of aromatic amino acids and metabolism of alanine, aspartate, and glutamate were associated with CM.**

Supplemental Materials:

Supplemental Table S11. Pathway comparison of metabolites upregulated in CM compared to controls with uncomplicated malaria and no history of CM using MetaboAnalyst 5.0.

	Total	Expected	Hits	p	Impact
Phenylalanine, tyrosine, and tryptophan biosynthesis	4	0.12	2	4.77E-03	1
Phenylalanine metabolism	10	0.29	2	3.20E-02	0.36
Lysine degradation	25	0.73	3	3.37E-02	0.14
Alanine, aspartate, and glutamate metabolism	28	0.81	3	4.51E-02	0.05

• I think the novel data that they want to highlight is the role of metalloproteinases in the CNS, can they expand on this, what cell type is involved, how does this peripheral signature reflect CNS pathology-if this is not the main message, what is and can it be supported with some of the different datasets?

We thank the reviewer for this comment and have further elaborated on the role of metalloproteinases in CNS pathology in the following way (Discussion lines 213-231):

Matrix metalloproteinases (MMPs) are a family of zinc-dependent endopeptidases that play an essential role in the extracellular matrix breakdown, including the BBB in the central nervous system (PMID: 11687497, 25306400). Transcription of *MMP8* was increased in all SM subtypes in comparison to uncomplicated controls. *MMP8* encodes for matrix metalloproteinase 8, a neutrophil-secreted granule protein with collagenase activity previously associated with blood brain barrier breakdown in malarial retinopathy and CM^{12,13,14}. Genetic downregulation and pharmacologic inhibition of *MMP8* have been associated with neuroprotection in a murine sepsis model, where animals with elevated levels of the protein had increased central nervous system inflammation and toxicity¹⁵. *MMP9* encodes for a protein involved in digestion of gelatin, the denatured form of collagen, and is released by neutrophils, endothelial cells, and astrocytes (PMID: 30086289)¹⁶. Activation of *MMP9* leads to the digestion of type IV collagen, laminin and fibronectin, which in turn leads to the breakdown of BBB tight junctions (PMID: 38008261). *MMP9* transcripts were significantly associated with CM and concurrent CM and SMA, suggesting that they might be specific for SM subsets that involve cerebral disease. Tissue inhibitor of metalloproteinases-1 (TIMP1) is known to protect the BBB by inhibition of active MMPs and has neuroprotective effects through dampening the effects of glutamate in excitotoxic injury (PMID: 12595242). TIMP1 has been shown to bind *MMP9*, and it is secreted by many tissues, including astrocytes in the central nervous system (PMID: 14648584, 7674941). TIMP1 had the strongest association in cases of CM compared to uncomplicated controls in proteomic analyses, and it was upregulated in CM. TIMP1 was also upregulated in CM cases compared to uncomplicated controls in transcriptomic analyses (logFC 0.76), but the p-value was not significant after correcting for multiple testing (unadjusted 0.012, adjusted 0.11). Evaluation of serum levels of *MMP8* and TIMP1 in children from Gabon has shown that TIMP1 was associated with signs and symptoms of severe malaria, whereas *MMP8* was elevated in both severe malaria and unmatched mild malaria cases compared to healthy controls¹⁷. In this study, *MMP8* and *MMP9* associated with SM subtypes compared to uncomplicated controls, but they were not differentially expressed in direct comparisons of SM subtypes with each other. Power for the latter analyses may have been limited due to sample sizes and a lack of matching.

• Can they integrate the metabolomics data with the transcriptional data (particularly where they identify differential metabolite pathways). Is a main finding from the abstract L-arginine metabolites were decreased in cerebral malaria, which coupled with increased ARG1 transcription suggests a putative mechanism impairing cerebral vasodilation. (ie they highlight a main finding in the abstract).

We thank the reviewer for this comment and we have further elaborated on the evidence behind this purported mechanism in the following manner (Discussion, lines 317-330):

Metabolomic analysis demonstrated significantly decreased levels of L-arginine in CM compared to uncomplicated malaria controls, which supports the previously established association between low serum L-arginine levels and CM^{58,59}. L-arginine is a substrate for nitric oxide production through the action of nitric oxide synthase. L-arginine depletion, and subsequent inability to produce nitric oxide,

likely leads to impaired vasodilation, which is part of the pathogenic cascade observed in CM⁶⁰. Arginase-1, encoded by *ARG1*, catalyzes the hydrolysis of L-arginine to L-ornithine and urea as part of the urea cycle. In our analyses, *ARG1* was expressed at higher levels in all subtypes of SM, including the corresponding transcriptomic comparison of CM to uncomplicated controls without a history of CM. Previous studies have suggested that low serum arginine results from limited bioavailability of glutamine and proline, which are used to synthesize arginine⁶¹. Our findings suggest that L-arginine depletion may also result from increased levels of arginase-1, **which is a putative mechanism for impaired cerebral vasodilation observed in CM. Furthermore**, rat models of acute pulmonary embolism demonstrated that L-arginine attenuates MMP9 activity through increased nitric oxide synthesis⁶². Cell culture experiments have also demonstrated that low nitric oxide concentrations lead to increased enzymatic activity of MMP9 and suppression of exogenous TIMP1, while high nitric oxide levels attenuate MMP9 activity⁶³.

- *They did not have protein data for the top associating transcripts from their analysis. If they could confirm protein differences for some of the transcripts (the ones that may inform their new disease a model information for example) (ie they highlight a main finding in the abstract “Tissue inhibitor of metalloproteinases 1 was the most upregulated protein in cerebral malaria, which along with elevated MMP8 and MMP9 transcription, underscores the importance of the metalloproteinase pathway” can any of these proteins differentials be confirmed?*

We thank the reviewer for this comment. While we addressed the limitations of an agnostic, mass spectroscopy-based proteomic approach in lines 355-358 in the discussion, this merits further elaboration. We have added to the Discussion to highlight published evidence of the requested associations:

While our approach was more tailored for hypothesis generation, it limited our analysis in that we had transcriptome data for only six of the 180 proteins assessed in the proteomic analyses, and we did not have protein data for the top associating transcripts. Replication of the metabolite and protein associations with CM via targeted approaches is essential toward producing generalizable results, and will be included in planned follow-up studies of severe malaria. **Of note, our transcriptomic findings replicate published associations of MMP8 and MMP9 protein with CM as well as an established association of TIMP1 transcript levels in an experimental murine cerebral malaria (PMID: 24467887, 16865090)^{14,15}.**

- *Table 1, as ethnicity is not compared between patients and most are Dogon, no need to specify their ethnicity; same with blood type, unless the comparison between blood type was done, or would have a major effect on one of the results. Same with Rh factor.*

We thank the reviewer for this comment and we have revised Table 1, and Supplemental Tables S3, S4, S7, and S12, accordingly.